# Instant Video Models: Universal Adapters for Stabilizing Image-Based Networks

**Matthew Dutson, Nathan Labiosa, Yin Li, and Mohit Gupta**
University of Wisconsin–Madison
{dutson,nlabiosa,yin.li,mgupta37}@wisc.edu

## Abstract

When applied sequentially to video, frame-based networks often exhibit temporal inconsistency—for example, outputs that flicker between frames. This problem is amplified when the network inputs contain time-varying corruptions. In this work, we introduce a general approach for adapting frame-based models for stable and robust inference on video. We describe a class of *stability adapters* that can be inserted into virtually any architecture and a resource-efficient training process that can be performed with a frozen base network. We introduce a unified conceptual framework for describing temporal stability and corruption robustness, centered on a proposed accuracy-stability-robustness loss. By analyzing the theoretical properties of this loss, we identify the conditions where it produces well-behaved stabilizer training. Our experiments validate our approach on several vision tasks including denoising (NAFNet), image enhancement (HDRNet), monocular depth (Depth Anything v2), and semantic segmentation (DeepLabv3+). Our method improves temporal stability and robustness against a range of image corruptions (including compression artifacts, noise, and adverse weather), while preserving or improving the quality of predictions.

## 1   Introduction

Video is often processed *frame-wise*—meaning images are passed one by one to a processing pipeline or model, and each output is independent of the previous one. This design choice is often driven by practical considerations: single-frame datasets are generally more diverse and accessible than video datasets, training image-based models is far less demanding in terms of compute and memory, and improvements in single-frame performance often carry over to video-based tasks.

Unfortunately, frame-wise processing faces the inherent challenge of *temporal consistency*, where model predictions fluctuate over time (Figure 1 top-middle). This behavior is especially problematic in tasks such as denoising or stylization, where temporal inconsistency can significantly reduce perceptual quality. Even in applications where the model output is not intended for human consumption, instability can impact downstream tasks. For example, inconsistent monocular depth estimates could produce erratic behavior in a collision avoidance system. Moreover, instability can reduce perceived reliability and thereby undermine user trust, regardless of objective performance.

Temporal consistency is closely related to *corruption robustness*. In field deployments, vision systems often operate in non-ideal conditions. For example, an autonomous vehicle may encounter inclement weather, sensor artifacts, or low-light noise; robustness in these circumstances is vital for safe system operation. These real-world corruptions are often *transient*, with an appearance that changes between frames (Figure 1 top-right). While it may be challenging to correct such degradations with a single frame, we can often infer the underlying clean signal from recent context. In this case, we can view robustness as a natural extension of temporal consistency.

39th Conference on Neural Information Processing Systems (NeurIPS 2025).

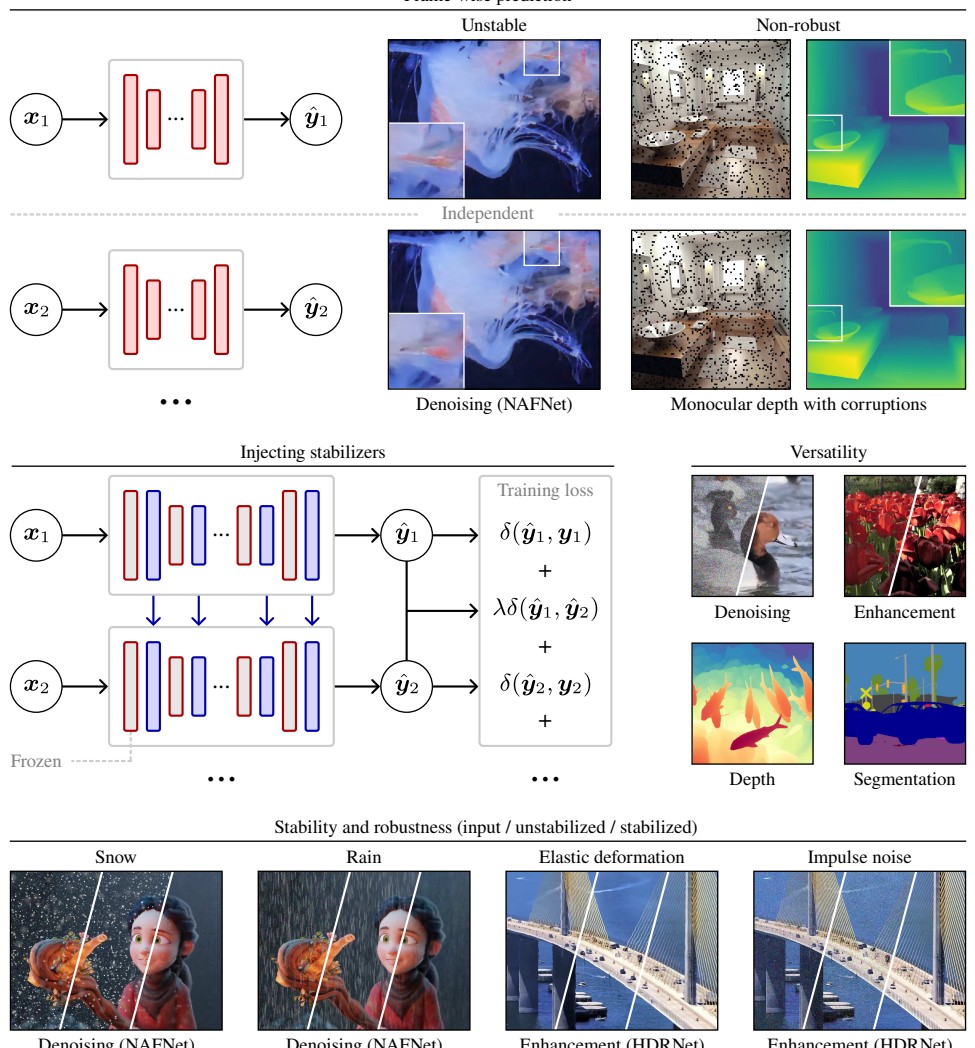

Figure 1: **Stabilizing image-based networks. (top)** Applying single-image models sequentially to the frames of a video can cause unstable predictions and failures under time-varying corruptions. In the top-right example, we see that randomly dropping patches causes artifacts in monocular depth estimates. **(middle)** We propose a method for injecting stabilizers into existing networks and for training these stabilizers using a unified accuracy-stability-robustness loss. **(bottom)** We demonstrate improvements in stability and robustness for various tasks, without modifying the original image-based models. Image sources: [18, 56, 58, 75].

Several prior works have proposed video-centric models with improved temporal consistency [5, 44, 24, 74, 60, 72]. However, these methods are often narrowly designed for one or a few tasks and require costly training on large-scale video datasets. Consequently, they lack the flexibility to leverage the extensive ecosystem of frame-based imaging and perception models. Further, few explicitly address robustness to transient corruptions or other challenging conditions.

In this work, we improve the temporal consistency and robustness of *pre-trained, image-based* models across various tasks. One of our primary challenges is that increasing stability may result in over-smoothing, which can, in turn, reduce accuracy. We conceptually explore the tradeoffs between output quality, corruption robustness, and temporal consistency, and introduce a unified accuracy-robustness-stability loss to balance these objectives. We provide a theoretical analysis of this loss and identify strategies to avoid "over-smoothing reality," such that there is no incentive for predictions to be smoother than the true scene dynamics.

Guided by this analysis, we propose a class of versatile *stabilization adapters* (Figure 1 middle). These adapters generate control signals, based on recent spatiotemporal context, that modulate changes to the model's features and output. By operating in both the feature and output spaces, we allow the adapters to model stability wherever it exists in the visual hierarchy. This property is important for high-level vision tasks, where stability is often best described in a feature space.

Our method offers several key benefits. First, our stabilization adapters are lightweight and modular, and do not require modifying the original model parameters. Second, our adapters operate causally; stabilized outputs depend only on current and past inputs—a feature that is critical for processing streaming video in latency-sensitive applications. Third, our approach is compatible with both low-level tasks, where stability can be described in terms of pixel values, and higher-level tasks, where stability occurs at the level of scene semantics. Finally, our method naturally enhances robustness to transient corruptions, without requiring explicit corruption modeling.

We evaluate our method on a range of tasks: denoising, image enhancement, monocular depth estimation, and semantic segmentation (Figure 1 middle-right). We also demonstrate improved robustness against various transient corruptions, including noise, dropped patches, elastic deformations, compression artifacts, and adverse weather (Figure 1 bottom). In most cases, these improvements do not reduce accuracy—on the contrary, we often see significant improvements in task metrics. Overall, our experiments establish the flexibility and practicality of our approach.

## 2 Related Work

**Corruption robustness.** Several prior works have addressed robustness against input corruptions. Hendrycks and Dietterich [23] propose metrics for measuring the robustness of image classifiers against common corruptions (e.g., compression artifacts or weather); their metrics inspire our definitions in Section 3. In general, natural corruptions have received less attention [14] from the vision community than adversarial corruptions [1, 19, 46, 48, 59, 61, 69, 71], although a handful of methods and benchmarks exist [3, 34, 41, 47, 70]. Like these works, our paper emphasizes robustness to naturally occurring corruptions rather than worst-case adversarial perturbations.

**Consistent image enhancement.** Frame-to-frame flickering is a significant problem for low-level image enhancement models; as such, there have been several works that improve temporal consistency for these tasks [5, 33, 37, 77, 80]. Blind video temporal consistency methods [5, 33, 37] treat the frame-level model as a black box, which allows generalization across models and applications. However, because they consider only the model input and output, these methods cannot model higher-level (semantic) stability and are prone to instability when the input is impacted by transient noise or corruptions. In contrast, we model stability in both the output space and the feature space, improving robustness against corrupted inputs and supporting a wider range of tasks, including those where stability is best described in semantic terms.

**Task-specific video architectures.** Many works propose video-optimized architectures for specific tasks, with temporal consistency a stated priority in many cases. This problem has been widely studied for ill-posed, low-level tasks where the output is intended for a human viewer; examples include video colorization [36, 44, 64, 76, 79, 82, 83], stylization [7, 10, 11, 13, 16, 17, 21, 24, 39, 43, 53, 57, 67, 73], and inpainting [6, 30, 35, 65, 74, 78]. Temporal consistency is also a concern in higher-level tasks, including segmentation [2, 51, 55, 60, 66], object detection [4, 26, 42, 63, 72, 84], and depth

estimation [27, 28, 32, 38, 45, 50, 68, 81]. Clockwork ConvNets [60] leverage the observation that semantic content evolves more slowly and smoothly than pixel values; we share this motivation in designing feature-domain stabilizers.

Our goal is not to design a task-specific method; in fact, we expect specialized architectures to outperform our general approach on benchmarks. The appeal of our approach lies in its practicality and versatility. Our method requires minimal training and no alterations to the original network, and can be applied to a broad range of video inference tasks, including those where highly optimized video architectures may not exist.

## 3 Defining Stability and Robustness

We start by defining temporal stability. Let $f_\phi : \mathcal{X} \to \mathcal{Y}$ be a frame-wise predictor with parameters $\phi$, and let $\delta(\boldsymbol{y}_1, \boldsymbol{y}_2)$ be a metric defined on the space $\mathcal{Y}$. We use $\hat{\boldsymbol{y}}$ to indicate the model output and $\boldsymbol{y}$ for the target output (ground truth, or features from a reference model). We formulate our definition as an expectation over data distribution $\mathcal{D}$ containing $(\boldsymbol{x}, \boldsymbol{y})$ sequences of duration $\tau$ indexed by discrete time step $t$ (the frame index). We define the stability $\mathcal{S}$ as the negative expected difference between adjacent predictions, i.e.,

$$\mathcal{S} = -\mathbb{E}_{(\boldsymbol{x}, \boldsymbol{y}, \tau) \sim \mathcal{D}} \left[ \sum_{t=1}^{\tau-1} \delta(f_\phi(\boldsymbol{x}_t), f_\phi(\boldsymbol{x}_{t+1})) \right]. \tag{1}$$

The negation is added such that stability increases as frame-to-frame variation decreases.

This notion of stability is closely related to robustness, i.e., the correctness of the model predictions under input corruptions [23]. The same input corruptions can cause both temporal instability and reduced prediction accuracy; examples include sensor noise, image or video compression artifacts, rain, and snow. Hendrycks and Dietterich [23] define corruption robustness $\mathcal{R}_c$ as the expected accuracy of a classifier under a distribution $\mathcal{E}$ of per-image perturbation functions. We extend their definition to cover arbitrary metrics $\delta$ and time series of duration $\tau$,

$$\mathcal{R}_c = -\mathbb{E}_{\varepsilon \sim \mathcal{E}, (\boldsymbol{x}, \boldsymbol{y}, \tau) \sim \mathcal{D}} \left[ \sum_{t=1}^{\tau} \delta\left(f_\phi(\varepsilon_t(\boldsymbol{x}_t)), \boldsymbol{y}_t\right) \right], \tag{2}$$

where $\varepsilon_t$ is the per-frame perturbation at time $t$. Again, $\boldsymbol{y}_t$ is the target output.

We define the corruption stability $\mathcal{S}_c$ similarly:

$$\mathcal{S}_c = -\mathbb{E}_{\varepsilon \sim \mathcal{E}, (\boldsymbol{x}, \boldsymbol{y}, \tau) \sim \mathcal{D}} \left[ \sum_{t=1}^{\tau-1} \delta(f_\phi(\varepsilon_t(\boldsymbol{x}_t)), f_\phi(\varepsilon_{t+1}(\boldsymbol{x}_{t+1}))) \right]. \tag{3}$$

Both $\mathcal{R}_c$ and $\mathcal{S}_c$ include input perturbations $\varepsilon$. $\mathcal{R}_c$ measures how accurately a model $f$ predicts the target, while $\mathcal{S}_c$ captures the temporal smoothness of the model's outputs. Notably, $\mathcal{R}_c$ and $\mathcal{S}_c$ can be applied to both the intermediate features and the output of $f$.

## 4 Learning to Balance Stability and Robustness

We now combine $\mathcal{R}_c$ and $\mathcal{S}_c$ to form a unified accuracy-stability-robustness training loss $\mathcal{U}_c$, and analyze the conditions under which this loss leads to well-behaved training.

**Unified accuracy-stability-robustness loss.** We define the unified loss $\mathcal{U}_c$ as

$$\mathcal{U}_c = -(\mathcal{R}_c + \lambda \mathcal{S}_c) \tag{4}$$

$$= \mathbb{E}_{\varepsilon \sim \mathcal{E}, (\boldsymbol{x}, \boldsymbol{y}, \tau) \sim \mathcal{D}} \left[ \sum_{t=1}^{\tau} \delta(f_\phi(\varepsilon_t(\boldsymbol{x}_t)), \boldsymbol{y}_t) + \lambda \sum_{t=1}^{\tau-1} \delta(f_\phi(\varepsilon_t(\boldsymbol{x}_t)), f_\phi(\varepsilon_{t+1}(\boldsymbol{x}_{t+1}))) \right], \tag{5}$$

where $\lambda$ is a constant that weights stability relative to accuracy.

**Theoretical analysis.** If $\delta$ can be expressed in terms of a norm on $\mathcal{Y}$, we can derive two bounds on $\lambda$. The first, $\lambda < 1/2$, which we call the *oracle bound*, defines the range of $\lambda$ where the ground truth is

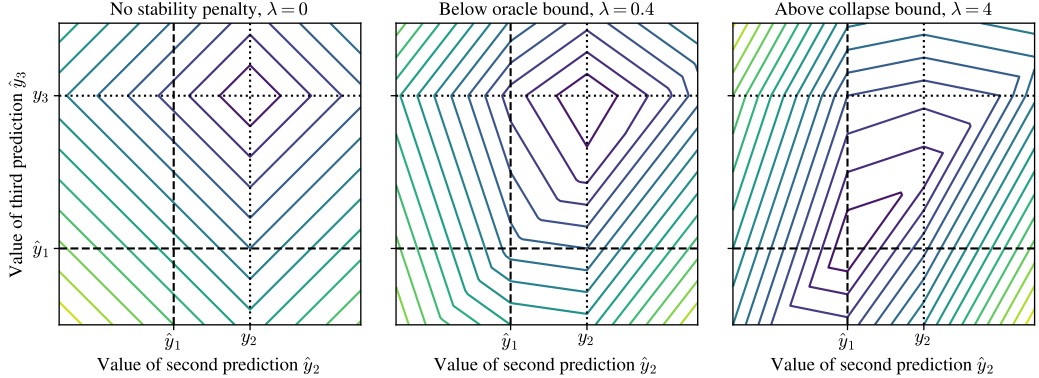

Figure 2: **Unified loss for one-dimensional predictions.** We consider a time series of duration $\tau = 3$ consisting of one-dimensional predictions, with $\delta$ defined as the L1 distance. We assume that the first prediction $\hat{y}_1$ is fixed (cannot be modified by a stabilization adapter). We show the value of the second prediction $\hat{y}_2$ along the x-axis and the value of the third $\hat{y}_3$ along the y-axis, with contours indicating the value of the unified loss as these predictions vary. When $\lambda = 0$, the minimum occurs at the ground truth $\hat{y}_2 = y_2$ and $\hat{y}_3 = y_3$. When $\lambda$ is nonzero but below the oracle bound, the minimum still occurs at the ground truth, but the loss increases more slowly in the direction of stabler predictions. When $\lambda$ exceeds the collapse bound, the global minimum is the collapse state $\hat{y}_3 = \hat{y}_2 = y_1$.

the global minimizer of the loss in prediction space. Under this bound, a perfectly accurate (oracle) model will never have an incentive to diverge from the correct prediction to increase stability. For each training item $x$, the minimum loss occurs at the ground truth $y$, implying zero gradients with respect to the prediction $\hat{y}$. See Appendix A for a proof of this and the following bound.

The second bound, $\lambda > \tau - 1$, is the *collapse bound*, and gives the range of $\lambda$ where the global loss minimizer corresponds to exact repetition of the initial prediction, regardless of scene changes. Unlike the global accuracy minimizer (the oracle state), which may be difficult or impossible to reach with gradient descent, the collapse state is often easily achieved. For example, if there is an EMA stabilizer (Section 5) on the output, we can achieve collapse simply by setting its decay to zero. In our experiments, we confirm that setting $\lambda$ above the collapse bound leads to prediction collapse, provided the collapse state is representable in the stabilizer parameter space.

**Example.** In Figure 2, we plot the loss as a function of two one-dimensional predictions $\hat{y}_2$ and $\hat{y}_3$, for several values of $\lambda$. When the stability penalty is introduced, the loss begins to tilt toward more stable predictions. When $\lambda$ is within the oracle bound, the global minimizer is unchanged. When it exceeds the collapse bound, the global minimizer is a repeated prediction.

Note that the oracle state is mutually exclusive with the collapse state (unless the ground truth is itself collapsed), as for any time series with $\tau > 1$, we have $\tau - 1 > 0.5$. Thus, we recommend training with $\lambda < 0.5$ in general; doing so ensures non-collapse and yields the correct behavior in the limiting case where the model is perfectly accurate.

## 5   Designing Stabilization Adapters

Our goal is to improve the temporal stability and robustness of a pre-trained, frame-wise predictor $f$ for video tasks. We assume $f$ is realized using a deep neural network with pre-trained weights $\phi_0$. While the unified loss $\mathcal{U}_c$ allows us to update $\phi_0$, this fine-tuning may be computationally expensive or require substantial training data. Instead, we consider adaptation of $f$, where a task-specific, lightweight adapter parametrized by $\Delta\phi$ is learned to stabilize the intermediate features and outputs of $f$. Under this formulation, we train only the parameters $\Delta\phi$ of the adapter (i.e., the stabilizer), and the original weights $\phi_0$ remain fixed.

**Design principles.** The following principles guide our stabilizer design. *First*, we consider only causal stabilizers, where the stabilized outputs at time $t$ are computed exclusively using information from times $\leq t$. This constraint is critical for processing streaming videos. *Second*, we stabilize both network activations (features) and outputs. Output-only stabilization is sufficient for some low-level

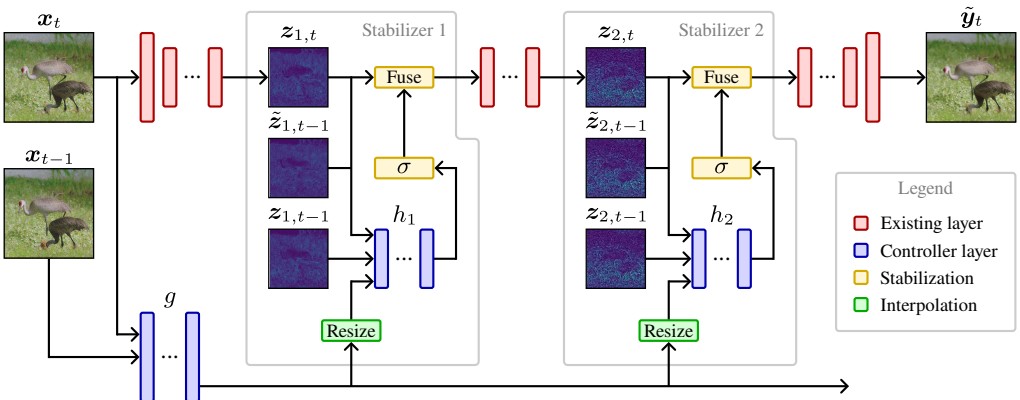

Figure 3: **Stabilization controllers.** Starting with the existing network (red), we add stabilizers (yellow) to select layers. The degree of stabilization, i.e., the decay $\beta$, can be predicted by a stabilization controller (blue). This controller consists of a shared backbone $g$ and one head $h_i$ per stabilized layer. Stabilizers can be added to both internal layers and the model output.

operations such as colorization. Feature-domain stabilization expands the potential scope of our method to include higher-level tasks; for these tasks, the inherent stability of a scene is often best described in feature space. *Finally*, we limit ourselves to designs that do not interfere with the existing network architecture. Our stabilizers are layer-level adapters with independent parameters and are designed to preserve the existing feature representation.

**Exponential moving average (EMA) stabilizer.** As a starting point, we consider a simple temporal smoothing operation applied to individual feature values forming a one-dimensional time series. Let $z_t$ denote an activation or feature value at time $t$, and let $\tilde{z}_t$ denote the corresponding stabilized activation. The exponential moving average (EMA) stabilizer produces a linear combination of the current unstabilized output and the previous stabilized output,

$$\tilde{z}_t = \beta z_t + (1 - \beta)\tilde{z}_{t-1}, \tag{6}$$

where $\beta \in [0, 1]$ is a decay-rate parameter. The recursive formulation here is equivalent to convolving the input time series with an infinite exponentially decaying weight kernel. The EMA stabilizer is memory-efficient (only $\tilde{z}_{t-1}$ is retained) and differentiable with respect to $\beta$ and $z$. We use the EMA stabilizer as the basis for more sophisticated designs.

**Stabilization controllers.** Simple smoothing operations (like the EMA stabilizer) are limited in both their spatial context and their ability to model complex changes in the scene. To address these limitations, we propose a *stabilization controller network*, which considers prior context (e.g., the current and previous input frames and feature maps) and predicts the amount of stabilization that should be applied to each value.

Although the idea of controller-augmented stabilization can be applied to many stabilization mechanisms, we focus here on the EMA stabilizer due to its differentiability and low memory requirements. In this case, the controller predicts the decay $\beta$ for each activation across layers. Our architecture, as shown in Figure 3, consists of a shared backbone $g$ and a stabilization head $h_i$ per stabilized layer. Tog $g$ compares the current and previous frames, offering a shortcut connection from frames to features. $h$ predicts the decay values based on the output of $g$ (resized to match the layer resolution), the current unstabilized features $z_t$, and the previous stabilized and unstabilized features $\tilde{z}_{t-1}$ and $z_{t-1}$. Together, the parameters of $g$ and $\{h_i\}$ form $\Delta\phi$.

Formally, the stabilized feature tensor $\tilde{z}_{i,t}$ for layer $i$ and time $t$ is given by

$$\tilde{z}_{i,t} = \beta_{i,t} \odot z_{i,t} + (1 - \beta_{i,t}) \odot \tilde{z}_{i,t-1}, \tag{7}$$

$$\beta_{i,t} = \sigma(h_i(g(x_t, x_{t-1}), z_{i,t}, \tilde{z}_{i,t-1}, z_{i,t-1})), \tag{8}$$

where $\odot$ denotes an element-wise product and $z_{i,t}$ is the unstabilized feature tensor.

We note that our controller is conceptually similar to selective state space models [20], although the design differs significantly. Equation 7 can be viewed as a linear dynamical system defined on frame-level features with parameters conditioned on the input, current, and previous features.

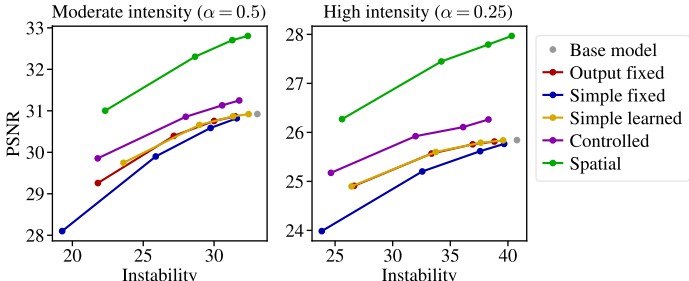

Figure 4: **Image enhancement results.** Introducing a controller and spatial fusion to the stabilizer significantly improves the accuracy-stability tradeoff. The spatial-fusion stabilizer reduces frame-to-frame variation by up to $\approx 35\%$ while exceeding the quality of the base model. "Instability" here refers to negative stability $(-\mathcal{S})$; see Equation 1. The goal is to move toward $-x$ (lower instability) and $+y$ (better image quality).

**Controller with spatial fusion.** Often, $\boldsymbol{z}$ takes the form of a 2D feature map. If $g$ and $h$ are convolutional, the predicted decay $\beta$ is informed by the frames and features within a local receptive field. However, with Equation 8, the weighted fusion used to compute $\tilde{\boldsymbol{z}}$ is still constrained to the time axis. Extending the weighted fusion to a spatial neighborhood can improve stabilization in the presence of motion by allowing translation of features from previous frames.

To perform spatial fusion, we modify the controller head $h$ to predict a spatial decay kernel $\boldsymbol{\eta}$ at each pixel rather than a single decay $\beta$. For a neighborhood that contains $m$ locations (including the central pixel), the kernel $\boldsymbol{\eta}$ contains $m + 1$ elements. The first $m$ elements weight the stabilized activations from the previous time step ($\tilde{\boldsymbol{z}}_{t-1}$), for each location in the neighborhood. The $(m + 1)$th element weights the current unstabilized activation ($\boldsymbol{z}_t$) for the central pixel. The kernel is softmax-normalized. The first $m$ logits are predicted directly by the controller head, and the last is set to zero (when $m = 1$, the softmax reduces to the sigmoid in Equation 8). See Section D.3 for more details.

The spatial fusion stabilizer can represent a recursive shift projection, where a feature vector is translated on each frame by an amount corresponding to the object motion. The maximum trackable motion in this case is determined by the spatial extent of the kernel $\boldsymbol{\eta}$.

## 6 Experiments

In Sections 6.1 and 6.2, we test our approach on image enhancement and denoising, respectively. We ablate the components of our method and explore the tradeoff between stability and accuracy. In Section 6.3, we test our stabilizers in the presence of various image corruptions, evaluating image enhancement, denoising, and depth estimation. In Section 6.4, we consider corruptions resulting from adverse weather (rain and snow).

Some low-level details (e.g., training hyperparameters) are omitted here for brevity; see the appendices for a more exhaustive description of experiment protocols. The appendices also include results for semantic segmentation and an exploration of training-free stabilizer composition.

**Variants and baselines.** Across our experiments, we consider a common set of variations (ablations of our approach) and baselines. The *simple learned* variant adds a simple EMA stabilizer (Equation 6) to the output and features, with one learned $\beta$ value per channel. The *controlled* variant adds a learned controller that predicts the stabilizer decay according to Equation 8. Finally, the *spatial* variation augments the controlled stabilizer by adding spatial fusion. As for baselines: the *simple fixed* method applies a simple EMA stabilizer to internal features and the output, with one global, hand-tuned $\beta$; the *output fixed* method applies a hand-tuned EMA stabilizer only to the model output.

### 6.1 Image Enhancement

**Task, dataset, and base model.** We first consider image enhancement, a task where perceptual quality (including stability) is of primary importance. We use the HDRNet model [18], which can be trained to reproduce many low-level image transformations. Specifically, we target the local

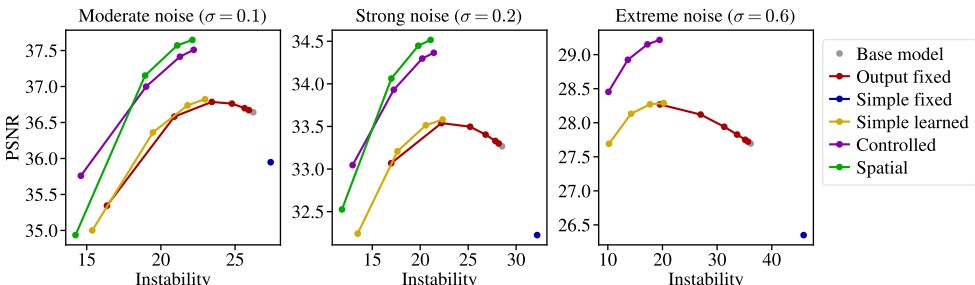

Figure 5: **Denoising results.** Because it attempts to stabilize an iid noise residual, naive feature-space stabilization leads to worse PSNR *and* worse stability. We achieve the best performance with a controlled stabilizer, usually with spatial fusion.

Laplacian detail-enhancement operator [49], due to its known forward model and readily available code. We consider two effect strengths: moderate ($\sigma = 0.4$ and $\alpha = 0.5$) and strong ($\sigma = 0.4$ and $\alpha = 0.25$). We generate training pairs by applying the local Laplacian filter to each frame of the Need for Speed (NFS) dataset [29]. NFS contains 100 videos (380k frames) collected at 240 FPS; we randomly select 20 videos for validation and use the remaining 80 for training. Videos are scaled to have a short-edge length of 360. We evaluate PSNR and instability (Equation 1) with $\delta = || \cdot ||_2$.

**Experiment protocol.** We fine-tune the original HDRNet `local_laplacian/strong_1024` weights for both effect strengths. After this fine-tuning, we attach a stabilizer to the output of each convolution and the overall model output (because the HDRNet architecture is extremely lightweight, this does not represent an unreasonable overhead). We then freeze the fine-tuned weights and train the stabilizers using BPTT on short video snippets ($\tau = 8$). We use the unified loss with $\delta = || \cdot ||_2$. We test several $\lambda$ values—0.1, 0.2, 0.4, 0.8, and 8.0—for each effect strength and model variation. The first three values are within the oracle bound, and the last exceeds the collapse bound.

**Results.** Figure 4 illustrates how PSNR and instability change as we vary the degree of stabilization ($\lambda$ for learned stabilizers, $\beta$ for hand-tuned stabilizers). We observe that for static (non-controlled) stabilizers, there is no benefit to stabilizing in feature space. For the static methods (output fixed, simple fixed, and simple learned), increasing stability brings a reduction in quality as we begin to over-smooth the output. In contrast, for the controlled and spatial stabilizers, there is a region where both PSNR and stability are improved over the base model. Spatial fusion gives a significant improvement in output quality—roughly 2 dB for the high-intensity effect. We suspect this is related to the nature of the detail-enhancement task (its sensitivity to small-scale motion). Finally, we confirm that setting $\lambda = 8 > \tau - 1$ leads to prediction collapse (instability $< 10^{-3}$, indicating a constant prediction). This result confirms that the global collapse minimum is easily reached, despite the non-convex nature of the optimization in general.

## 6.2 Denoising

**Task, dataset, and base model.** Next, we evaluate image denoising under AWGN. Denoising highlights the utility of our method when the input is itself unstable. We use the NAFNet model [9], which employs a U-Net architecture with modified convolutional blocks (NAFBlocks). We again use the NFS dataset, with the same train/validation split as in Section 6.1. We evaluate three noise levels: moderate ($\sigma = 0.1$, for float images $\in [0, 1]$), strong ($\sigma = 0.2$), and extreme ($\sigma = 0.6$). See Appendix F.4 for additional results on the DAVIS [52] dataset.

**Experiment protocol.** We start by fine-tuning the unstabilized model for each dataset and noise level, initializing with the `nafnet_sidd_width32` weights published by the model authors. We then attach a stabilizer to the output of each NAFBlock and to the model output. As before, we freeze the fine-tuned weights and train only the stabilizer parameters, using the unified loss with $\delta = || \cdot ||_2$ and sweeping out $\lambda = 0.1, 0.2, 0.4, 0.8,$ and 8.0.

**Results.** Figure 5 shows PSNR and instability across noise levels. The "simple fixed" stabilizer gives a somewhat surprising result: adding stabilization worsens both PSNR and instability. The reason is that the network backbone predicts a noise residual, which is completely uncorrelated

| Corruption | Method | Enhancement | | Denoising | | Depth estimation | | |
|---|---|---|---|---|---|---|---|---|
| | | PSNR | Instability | PSNR | Instability | AbsRel ($\downarrow$) | Delta-1.25 ($\uparrow$) | Instability |
| Patch drop | Base model | 17.43 | 164.6 | 18.93 | 151.4 | 0.070 | 0.948 | 9.89 |
| | Ours | 31.39 | 30.36 | 35.46 | 20.42 | 0.070 | 0.956 | 4.73 |
| Elastic distortion | Base model | 24.00 | 64.23 | 27.99 | 47.31 | 0.052 | 0.968 | 7.76 |
| | Ours | 26.63 | 24.97 | 30.78 | 21.04 | 0.057 | 0.967 | 4.75 |
| Frame drop | Base model | 28.65 | 94.04 | 33.42 | 113.5 | 0.065 | 0.936 | 14.17 |
| | Ours | 31.69 | 29.14 | 27.34 | 20.56 | 0.050 | 0.974 | 4.91 |
| JPEG artifacts | Base model | 24.85 | 42.06 | 29.01 | 39.71 | 0.057 | 0.964 | 7.32 |
| | Ours | 26.46 | 23.58 | 32.19 | 20.49 | 0.065 | 0.961 | 4.92 |
| Impulse noise | Base model | 14.28 | 217.8 | 24.65 | 62.10 | 0.047 | 0.974 | 6.83 |
| | Ours | 27.37 | 30.90 | 32.04 | 19.73 | 0.056 | 0.972 | 4.98 |

Table 1: **Corruption robustness.** Our method significantly improves stability while preserving or improving prediction quality. The only notable exception is reduced PSNR for denoising with dropped frames. However, we note that PSNR (averaged over frames) does not capture the jarring perceptual effect of dropped frames; thus, the high PSNR of the base model is somewhat deceptive.

between frames. Over-smoothing this residual inhibits the noise removal, which worsens PSNR and increases frame-to-frame variation due to unremoved noise. Fortunately, this behavior does not exist for the learned or controlled stabilizers, as they target only those features that exhibit some temporal smoothness. Controlled stabilizers improve both stability and PSNR when $\lambda \leq 0.4$. We again confirm that setting $\lambda = 8 > \tau - 1$ leads to prediction collapse, i.e., instability $< 10^{-3}$.

In most cases, we find that the spatial fusion stabilizer outperforms the non-spatial controlled stabilizer. However, there is a notable exception in the case of extreme noise, where the spatial fusion stabilizer is about 6 dB worse than other methods (these points are outside the range of the plot in Figure 5 but are included in Table 7). Intriguingly, this quality gap only appears when evaluating long sequences (hundreds of frames) and shrinks as we reduce $\tau$. In Appendix F.6, we show that this problem can be at least partially mitigated by increasing $\tau$ during training.

The supplementary material includes video files comparing the stabilized and unstabilized outputs. We encourage the reader to watch these videos for a clear qualitative comparison (temporal inconsistency is much more obvious in videos than in side-by-side static frames).

### 6.3 Corruption Robustness

**Tasks, datasets, and base models.** We again consider HDRNet for image enhancement and NAFNet for denoising. We also include results for depth estimation with Depth Anything v2 [75]. Depth Anything is notable for the scale of data used in its training; it would be costly to develop a new video architecture from scratch, making Depth Anything a good candidate for our approach. See Appendix F.2 for results on semantic segmentation with DeepLabv3+.

For depth training, we use the VisionSim framework [25] (Blender) to generate a dataset of simulated videos with ground-truth depth. The dataset consists of 50 indoor scenes containing ego motion and is rendered at 50 FPS. We randomly select 10 scenes for validation and use the rest for training. See Appendix E for further details on this dataset.

**Experiment protocol.** For HDRNet and NAFNet, we train spatial-fusion stabilizers using the same settings as in Sections 6.1 and 6.2. We train HDRNet for the moderate effect strength ($\alpha = 0.5$), and NAFNet for moderate noise ($\sigma = 0.1$) on NFS. Unlike other models, we do not fine-tune the base Depth Anything model; we found that naive fine-tuning on a small dataset like ours quickly led to overfitting. We add controlled stabilizers to instances of `DepthAnythingReassembleLayer`, `DepthAnythingFeatureFusionLayer`, and the model output.

We train and evaluate stabilized models for each of the following corruptions: (1) randomly zeroing each $8\times8$ patch with probability 0.1, (2) elastic deformation, see Appendix E for details, (3) randomly zeroing each frame with probability 0.1, (4) applying JPEG compression at quality 10/100, and (5) adding impulse noise to each channel with probability 0.05 for both salt and pepper. We set $\lambda = 0.2$ when training stabilizers for corruption robustness.

| Stabilized? | Unfrozen? | Rain | | | Snow | | |
|---|---|---|---|---|---|---|---|
| | | PSNR | SSIM | Instability | PSNR | SSIM | Instability |
| × | × | 21.43 | 0.617 | 151.76 | 18.62 | 0.577 | 262.48 |
| ✓ | × | 28.63 | 0.880 | 57.88 | 31.34 | 0.914 | 59.31 |
| × | ✓ | 32.19 | 0.937 | 70.84 | 34.33 | 0.950 | 66.57 |
| ✓ | ✓ | 32.61 | 0.938 | 58.30 | 35.20 | 0.956 | 58.98 |

Table 2: **Adverse weather robustness on RobustSPRING.** Training stabilizers with a frozen base model gives substantial improvement over the unstabilized model. We obtain the best overall results by jointly training stabilizers with the base parameters.

**Results.** Table 1 shows accuracy and stability with and without stabilizers. Adding stabilizers leads to significant reductions in instability and, in most cases, improvements in per-frame accuracy metrics. See Appendix Figures 10, 11, and 12 for qualitative results.

### 6.4  Adverse Weather Robustness

**Task, dataset, and base model.** We now evaluate robustness under adverse weather conditions. Specifically, we consider the rain and snow corruptions from the RobustSpring [58] dataset. These corruptions differ from those in Section 6.3 in their spatial complexity and temporal dynamics (raindrops and snowflakes follow continuous paths, unlike simpler corruptions such as randomly-dropped patches). RobustSpring contains 10 rendered sequences (2000 total frames), each with left- and right-frame variants. Clean ground-truth frames are provided for all sequences. We randomly select 2 videos for validation and use the remaining 8 for training. Videos are downsized to 720p. We evaluate NAFNet denoising [9] with $\sigma = 0.1$, adding noise after weather effects.

**Experiment protocol.** We fine-tune the original `nafnet_sidd_width32` weights with noise, but not weather corruptions. Following Section 6.2, we then append a spatial-fusion stabilizer to each NAFBlock and the model output. We train these stabilizers under noise + weather using $\lambda = 0.2$. Consistent with our other experiments, we train stabilizers with a frozen base model.

In addition, we consider two variants with an unfrozen base model. In the first variant, we fine-tune the base model on noise + weather without stabilizers. In the other, we unfreeze the base model when training on noise + weather, jointly training the stabilizer and base parameters. For both variants, we use the same hyperparameters as the frozen stabilizer training.

**Results.** See Table 2 and Appendix Figure 14 for results. Compared to the unstabilized baseline, stabilizers substantially improve image quality and stability. Likewise, fine-tuning the original weights (without stabilizers) with weather corruptions gives a significant improvement. Compared to stabilization, fine-tuning gives better single-image quality but higher instability. We obtain the best overall results by combining fine-tuning with stabilization. In general, we expect this joint training approach to be the best choice for small- to medium-sized models. For larger models where training the base model is infeasible, training only the stabilizers still gives reasonable results.

## 7  Discussion

**Limitations.** The bounds in Section 4 assume that the distance $\delta$ can be expressed in terms of a norm on the prediction space $\mathcal{Y}$. This condition excludes many widely used loss functions, especially more sophisticated, multi-component losses. Nonconforming $\delta$ may still work in practice, although they may require more careful tuning of $\lambda$ due to the lack of theoretical guarantees.

We had some difficulty with sim-to-real generalization when training stabilizers for Depth Anything. We conjecture that real video contains subtle corruptions not present in simulated video—for example, sensor noise, compression artifacts, or optical phenomena. These "baseline corruptions" could explain some of the temporal instability we see when using apparently clean input video.

**Alternate metrics.** In all of our experiments, we use a simple Euclidean norm for $\delta$. However, other metrics may be a better choice, depending on the task. For example, a variant of the Wasserstein metric may give improved results for two-dimensional outputs and feature maps, due to its ability to account for the spatial structure of the tensor. See Appendix B for further discussion.

## Acknowledgments and Disclosure of Funding

Thanks to Sacha Jungerman for helping us generate VisionSim sequences for the depth estimation experiments. This research was supported by NSF CAREER award 1943149, NSF CPS grant 2333491, ARL under contract number W911NF-2020221, and the Wisconsin Alumni Research Foundation via a Research Forward Initiative.

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

# A  Proofs

This section contains proofs of the oracle and collapse bounds from Section 4.

## A.1  Oracle Bound

We assume that $\delta$ takes the form $\delta(\boldsymbol{a}, \boldsymbol{b}) = \zeta(\boldsymbol{a} - \boldsymbol{b})$ where $\zeta$ is a norm on $\mathbb{R}^d$. We define

$$u(\hat{\boldsymbol{y}}, \boldsymbol{y}) = \sum_{t=1}^{\tau} \delta(\hat{\boldsymbol{y}}_t, \boldsymbol{y}_t) + \lambda \sum_{t=1}^{\tau-1} \delta(\hat{\boldsymbol{y}}_t, \hat{\boldsymbol{y}}_{t+1}) \tag{9}$$

$$= \sum_{t=1}^{\tau} \zeta(\hat{\boldsymbol{y}}_t - \boldsymbol{y}_t) + \lambda \sum_{t=1}^{\tau-1} \zeta(\hat{\boldsymbol{y}}_t - \hat{\boldsymbol{y}}_{t+1}) \tag{10}$$

as the bracketed expression in Equation 4, using the shorthand $\hat{\boldsymbol{y}}_t = f(\varepsilon_t(\boldsymbol{x}_t))$. Our objective is to show that for any $\hat{\boldsymbol{y}} \neq \boldsymbol{y}$, there exists some $\boldsymbol{y}^\dagger \neq \hat{\boldsymbol{y}}$ such that $u(\boldsymbol{y}^\dagger, \boldsymbol{y}) < u(\hat{\boldsymbol{y}}, \boldsymbol{y})$.

There are two possible cases: (1) $\hat{\boldsymbol{y}}_t \neq \boldsymbol{y}_t$ for at least one of the endpoints, meaning $\hat{\boldsymbol{y}}_1 \neq \boldsymbol{y}_1$ or $\hat{\boldsymbol{y}}_\tau \neq \boldsymbol{y}_\tau$; and (2) $\hat{\boldsymbol{y}}_t = \boldsymbol{y}_t$ at both endpoints.

We start with the first case. We assume that $\hat{\boldsymbol{y}}_1 \neq \boldsymbol{y}_1$, with the proof being symmetric for $\hat{\boldsymbol{y}}_\tau \neq \boldsymbol{y}_\tau$. We propose $\boldsymbol{y}^\dagger$ where $\boldsymbol{y}_1^\dagger = \boldsymbol{y}_1$ and $\boldsymbol{y}_t^\dagger = \hat{\boldsymbol{y}}_t$ for $t > 1$. Starting with the definition of $u$,

$$u(\boldsymbol{y}^\dagger, \boldsymbol{y}) = \zeta(\boldsymbol{y}_1 - \boldsymbol{y}_1) + \lambda\zeta(\boldsymbol{y}_1 - \hat{\boldsymbol{y}}_2) + \ldots \tag{11}$$

$$= \lambda\zeta(\boldsymbol{y}_1 - \hat{\boldsymbol{y}}_2) + \ldots \tag{12}$$

$$u(\hat{\boldsymbol{y}}, \boldsymbol{y}) = \zeta(\hat{\boldsymbol{y}}_1 - \boldsymbol{y}_1) + \lambda\zeta(\hat{\boldsymbol{y}}_1 - \hat{\boldsymbol{y}}_2) + \ldots \tag{13}$$

By the triangle inequality and absolute homogeneity of $\zeta$,

$$u(\boldsymbol{y}^\dagger, \boldsymbol{y}) = \lambda\zeta(\boldsymbol{y}_1 - \hat{\boldsymbol{y}}_2) + \ldots \tag{14}$$

$$= \lambda\zeta((\boldsymbol{y}_1 - \hat{\boldsymbol{y}}_1) + (\hat{\boldsymbol{y}}_1 - \hat{\boldsymbol{y}}_2)) + \ldots \tag{15}$$

$$\leq \lambda\zeta(\boldsymbol{y}_1 - \hat{\boldsymbol{y}}_1) + \lambda\zeta(\hat{\boldsymbol{y}}_1 - \hat{\boldsymbol{y}}_2) + \ldots \tag{16}$$

$$= \lambda\zeta(\hat{\boldsymbol{y}}_1 - \boldsymbol{y}_1) + \lambda\zeta(\hat{\boldsymbol{y}}_1 - \hat{\boldsymbol{y}}_2) + \ldots \tag{17}$$

Assume that $\lambda < 1$. Then,

$$u(\boldsymbol{y}^\dagger, \boldsymbol{y}) \leq \lambda\zeta(\hat{\boldsymbol{y}}_1 - \boldsymbol{y}_1) + \lambda\zeta(\hat{\boldsymbol{y}}_1 - \hat{\boldsymbol{y}}_2) + \ldots \tag{18}$$

$$< \zeta(\hat{\boldsymbol{y}}_1 - \boldsymbol{y}_1) + \lambda\zeta(\hat{\boldsymbol{y}}_1 - \hat{\boldsymbol{y}}_2) + \ldots \tag{19}$$

$$= u(\hat{\boldsymbol{y}}, \boldsymbol{y}). \tag{20}$$

Therefore, $u(\boldsymbol{y}^\dagger, \boldsymbol{y}) < u(\hat{\boldsymbol{y}}, \boldsymbol{y})$.

To summarize, we have shown that if $\lambda < 1$, any $\hat{\boldsymbol{y}}$ where $\hat{\boldsymbol{y}}_1 \neq \boldsymbol{y}_1$ cannot minimize $u$. The same is true for $\hat{\boldsymbol{y}}_\tau \neq \boldsymbol{y}_\tau$.

Now we consider the second case, where $\hat{\boldsymbol{y}} \neq \boldsymbol{y}$ but their endpoints are the same. In this case, we must have $\hat{\boldsymbol{y}}_s \neq \boldsymbol{y}_s$ for some $1 < s < \tau$. We propose $\boldsymbol{y}_s^\dagger = \boldsymbol{y}_s$ and $\boldsymbol{y}_t^\dagger = \hat{\boldsymbol{y}}_t$ for $t \neq s$. Starting again with the definition of $u$,

$$u(\boldsymbol{y}^\dagger, \boldsymbol{y}) = \cdots + \lambda\zeta(\hat{\boldsymbol{y}}_{t-1} - \boldsymbol{y}_t) + \zeta(\boldsymbol{y}_t - \boldsymbol{y}_t) + \lambda\zeta(\boldsymbol{y}_t - \hat{\boldsymbol{y}}_{t+1}) + \ldots \tag{21}$$

$$= \cdots + \lambda\zeta(\hat{\boldsymbol{y}}_{t-1} - \boldsymbol{y}_t) + \lambda\zeta(\boldsymbol{y}_t - \hat{\boldsymbol{y}}_{t+1}) + \ldots \tag{22}$$

$$u(\hat{\boldsymbol{y}}, \boldsymbol{y}) = \cdots + \lambda\zeta(\hat{\boldsymbol{y}}_{t-1} - \hat{\boldsymbol{y}}_t) + \zeta(\hat{\boldsymbol{y}}_t - \boldsymbol{y}_t) + \lambda\zeta(\hat{\boldsymbol{y}}_t - \hat{\boldsymbol{y}}_{t+1}) + \ldots \tag{23}$$

By the triangle inequality and absolute homogeneity of $\zeta$,

$$u(\boldsymbol{y}^\dagger, \boldsymbol{y}) = \cdots + \lambda\zeta(\hat{\boldsymbol{y}}_{t-1} - \boldsymbol{y}_t) + \lambda\zeta(\boldsymbol{y}_t - \hat{\boldsymbol{y}}_{t+1}) + \ldots \tag{24}$$

$$= \cdots + \lambda\zeta((\hat{\boldsymbol{y}}_{t-1} - \hat{\boldsymbol{y}}_t) + (\hat{\boldsymbol{y}}_t - \boldsymbol{y}_t)) + \lambda\zeta((\boldsymbol{y}_t - \hat{\boldsymbol{y}}_t) + (\hat{\boldsymbol{y}}_t - \hat{\boldsymbol{y}}_{t+1})) + \ldots \tag{25}$$

$$\leq \cdots + \lambda\zeta(\hat{\boldsymbol{y}}_{t-1} - \hat{\boldsymbol{y}}_t) + \lambda\zeta(\hat{\boldsymbol{y}}_t - \boldsymbol{y}_t) + \lambda\zeta(\boldsymbol{y}_t - \hat{\boldsymbol{y}}_t) + \lambda\zeta(\hat{\boldsymbol{y}}_t - \hat{\boldsymbol{y}}_{t+1}) + \ldots \tag{26}$$

$$= \cdots + \lambda\zeta(\hat{\boldsymbol{y}}_{t-1} - \hat{\boldsymbol{y}}_t) + 2\lambda\zeta(\hat{\boldsymbol{y}}_t - \boldsymbol{y}_t) + \lambda\zeta(\hat{\boldsymbol{y}}_t - \hat{\boldsymbol{y}}_{t+1}) + \ldots \tag{27}$$

Assume that $\lambda < 1/2$. Then,

$$u(\boldsymbol{y}^\dagger, \boldsymbol{y}) \leq \cdots + \lambda\zeta(\hat{\boldsymbol{y}}_{t-1} - \hat{\boldsymbol{y}}_t) + 2\lambda\zeta(\hat{\boldsymbol{y}}_t - \boldsymbol{y}_t) + \lambda\zeta(\hat{\boldsymbol{y}}_t - \hat{\boldsymbol{y}}_{t+1}) + \dots \tag{28}$$

$$< \cdots + \lambda\zeta(\hat{\boldsymbol{y}}_{t-1} - \hat{\boldsymbol{y}}_t) + \zeta(\hat{\boldsymbol{y}}_t - \boldsymbol{y}_t) + \lambda\zeta(\hat{\boldsymbol{y}}_t - \hat{\boldsymbol{y}}_{t+1}) + \dots \tag{29}$$

$$= u(\hat{\boldsymbol{y}}, \boldsymbol{y}) \tag{30}$$

Therefore, $u(\boldsymbol{y}^\dagger, \boldsymbol{y}) < u(\hat{\boldsymbol{y}}, \boldsymbol{y})$.

We have now shown that if $\lambda < 1/2$, any $\hat{\boldsymbol{y}}$ where $\hat{\boldsymbol{y}}_s \neq \boldsymbol{y}_s$ for $1 < s < \tau$ cannot minimize $u$. Combining this with the result from the first case, we have shown that for $\lambda < 1/2$, $\hat{\boldsymbol{y}} = \boldsymbol{y}$ is the unique global minimizer of $u$. $\square$

## A.2 Collapse Bound

To prove the collapse bound, we first prove the convexity of $u$. We then show that, if $\hat{\boldsymbol{y}}_1$ is fixed (cannot be modified by the stabilizer), $\hat{\boldsymbol{y}}_s = \hat{\boldsymbol{y}}_1$ for $1 < s \leq \tau$ is a local minimizer of $u$ for $\lambda > \tau - 1$. Because $u$ is convex, this makes $\hat{\boldsymbol{y}}_s = \hat{\boldsymbol{y}}_1$ the global minimizer.

Define $\boldsymbol{q}$ as the concatenation $(\hat{\boldsymbol{y}}, \boldsymbol{y})$, i.e., the vector input to $u$. Consider two such inputs $\boldsymbol{q}$ and $\boldsymbol{q}^\dagger$. $u$ satisfies the triangle inequality;

$$u(\boldsymbol{q} + \boldsymbol{q}^\dagger) = \sum_{t=1}^{\tau} \zeta(\hat{\boldsymbol{y}}_t - \boldsymbol{y}_t + \hat{\boldsymbol{y}}_t^\dagger - \boldsymbol{y}_t^\dagger) + \lambda \sum_{t=1}^{\tau-1} \zeta(\hat{\boldsymbol{y}}_t - \hat{\boldsymbol{y}}_{t+1} + \hat{\boldsymbol{y}}_t^\dagger - \hat{\boldsymbol{y}}_{t+1}^\dagger) \tag{31}$$

$$\leq \sum_{t=1}^{\tau} \zeta(\hat{\boldsymbol{y}}_t - \boldsymbol{y}_t) + \lambda \sum_{t=1}^{\tau-1} \zeta(\hat{\boldsymbol{y}}_t - \hat{\boldsymbol{y}}_{t+1}) + \sum_{t=1}^{\tau} \zeta(\hat{\boldsymbol{y}}_t^\dagger - \boldsymbol{y}_t^\dagger) + \lambda \sum_{t=1}^{\tau-1} \zeta(\hat{\boldsymbol{y}}_t^\dagger - \hat{\boldsymbol{y}}_{t+1}^\dagger) \tag{32}$$

$$= u(\boldsymbol{q}) + u(\boldsymbol{q}^\dagger), \tag{33}$$

and is absolutely homogeneous;

$$u(c\boldsymbol{q}) = \sum_{t=1}^{\tau} \zeta(c\hat{\boldsymbol{y}}_t - c\boldsymbol{y}_t) + \lambda \sum_{t=1}^{\tau-1} \zeta(c\hat{\boldsymbol{y}}_t - c\hat{\boldsymbol{y}}_{t+1}) \tag{34}$$

$$= |c| \sum_{t=1}^{\tau} \zeta(\hat{\boldsymbol{y}}_t - \boldsymbol{y}_t) + |c|\lambda \sum_{t=1}^{\tau-1} \zeta(\hat{\boldsymbol{y}}_t - \hat{\boldsymbol{y}}_{t+1}) \tag{35}$$

$$= |c| u(\boldsymbol{q}), \tag{36}$$

implying $u$ is convex;

$$u(r\boldsymbol{q} + (1-r)\boldsymbol{q}^\dagger) \leq u(r\boldsymbol{q}) + u((1-r)\boldsymbol{q}^\dagger) \tag{37}$$

$$= |r| u(\boldsymbol{q}) + |1-r| u(\boldsymbol{q}^\dagger) \tag{38}$$

$$= r u(\boldsymbol{q}) + (1-r) u(\boldsymbol{q}^\dagger). \tag{39}$$

where $0 \leq r \leq 1$. Because $u$ is jointly convex on $\boldsymbol{q}$, it is also convex with respect to $(\hat{\boldsymbol{y}}_2, \dots, \hat{\boldsymbol{y}}_\tau)$ when $\boldsymbol{y}$ and $\hat{\boldsymbol{y}}_1$ are fixed.

Our goal is now to show that, assuming $\boldsymbol{y}$ and $\hat{\boldsymbol{y}}_1$ are fixed, $\hat{\boldsymbol{y}}_s = \hat{\boldsymbol{y}}_1$ for $1 < s \leq \tau$ is a local minimizer of $u$. That is, we want to find the $\lambda$ regime where any small movement away from this point increases the value of $u$. Assume a perturbed prediction

$$\boldsymbol{y}^\dagger = (\hat{\boldsymbol{y}}_1 + \boldsymbol{p}_1, \hat{\boldsymbol{y}}_1 + \boldsymbol{p}_2, \dots, \hat{\boldsymbol{y}}_1 + \boldsymbol{p}_\tau), \tag{40}$$

where $\boldsymbol{p}_1 = 0$. Starting with the definition of $u$,

$$u(\hat{\boldsymbol{y}}, \boldsymbol{y}) = \sum_{t=1}^{\tau} \zeta(\hat{\boldsymbol{y}}_1 - \boldsymbol{y}_t) + \lambda \sum_{t=1}^{\tau-1} \zeta(\hat{\boldsymbol{y}}_1 - \hat{\boldsymbol{y}}_1) \tag{41}$$

$$= \sum_{t=1}^{\tau} \zeta(\hat{\boldsymbol{y}}_1 - \boldsymbol{y}_t) \tag{42}$$

$$u(\boldsymbol{y}^{\dagger}, \boldsymbol{y}) = \sum_{t=1}^{\tau} \zeta(\hat{\boldsymbol{y}}_1 + \boldsymbol{p}_t - \boldsymbol{y}_t) + \lambda \sum_{t=1}^{\tau-1} \zeta((\hat{\boldsymbol{y}}_1 + \boldsymbol{p}_t) - (\hat{\boldsymbol{y}}_1 + \boldsymbol{p}_{t+1})) \tag{43}$$

$$= \sum_{t=1}^{\tau} \zeta(\hat{\boldsymbol{y}}_1 + \boldsymbol{p}_t - \boldsymbol{y}_t) + \lambda \sum_{t=1}^{\tau-1} \zeta(\boldsymbol{p}_t - \boldsymbol{p}_{t+1}) \tag{44}$$

Let

$$\theta = \underset{t \in 1:\tau}{\mathrm{argmax}} \zeta(\boldsymbol{p}_t) \tag{45}$$

be the time step with the largest-magnitude perturbation, and let $\phi = \zeta(\boldsymbol{p}_\theta)$ be the magnitude of this perturbation. Applying the reverse triangle inequality to the first summation in Equation 44 (the accuracy term), we have

$$\sum_{t=1}^{\tau} \zeta(\hat{\boldsymbol{y}}_1 + \boldsymbol{p}_t - \boldsymbol{y}_t) = \sum_{t=1}^{\tau} \zeta((\hat{\boldsymbol{y}}_1 - \boldsymbol{y}_t) - (-\boldsymbol{p}_t)) \tag{46}$$

$$\geq \sum_{t=1}^{\tau} [\zeta(\hat{\boldsymbol{y}}_1 - \boldsymbol{y}_t) - \zeta(-\boldsymbol{p}_t)] \tag{47}$$

$$= \sum_{t=1}^{\tau} \zeta(\hat{\boldsymbol{y}}_1 - \boldsymbol{y}_t) - \sum_{t=1}^{\tau} \zeta(\boldsymbol{p}_t) \tag{48}$$

$$= \sum_{t=1}^{\tau} \zeta(\hat{\boldsymbol{y}}_1 - \boldsymbol{y}_t) - \sum_{t=2}^{\tau} \zeta(\boldsymbol{p}_t) \tag{49}$$

$$\geq \sum_{t=1}^{\tau} \zeta(\hat{\boldsymbol{y}}_1 - \boldsymbol{y}_t) - (\tau - 1)\phi \tag{50}$$

So, introducing the perturbation reduces the accuracy term by at most $(\tau - 1)\phi$. Now considering the second summation (the stability term),

$$\lambda \sum_{t=1}^{\tau-1} \zeta(\boldsymbol{p}_t - \boldsymbol{p}_{t+1}) \geq \lambda \sum_{t=1}^{\theta} \zeta(\boldsymbol{p}_t - \boldsymbol{p}_{t+1}) \tag{51}$$

$$\geq \lambda \zeta(\boldsymbol{p}_1 - \boldsymbol{p}_\theta) \tag{52}$$

$$= \lambda \zeta(\boldsymbol{p}_\theta) \tag{53}$$

$$= \lambda \phi. \tag{54}$$

That is, introducing the perturbation increases the stability term by at least $\lambda \phi$.

Therefore, if $\lambda > \tau - 1$, the overall change in $u$ is positive, and we have shown that $\hat{\boldsymbol{y}}_s = \hat{\boldsymbol{y}}_1$ for $1 < s \leq \tau$ is a local minimizer of $u$. By convexity of $u$, this point is also a global minimizer. $\square$

## B  Transport Metric

In this section, we propose an alternate metric for use with our unified loss. Specifically, we describe a variant of the Wasserstein metric that accounts for the spatial structure of an image or feature tensor.

Let $\boldsymbol{z}_1, \boldsymbol{z}_2 \in \mathbb{R}^{h \times w}$ be two image or feature map channels, and let $\boldsymbol{a} = \boldsymbol{z}_1 - \boldsymbol{z}_2$. We define the transport distance $\mathcal{T}(\boldsymbol{a}) = \zeta(\boldsymbol{a}) = \delta(\boldsymbol{z}_1, \boldsymbol{z}_2)$ as the minimum cost of a linear optimization. Intuitively, this optimization finds the shortest correspondence from $\boldsymbol{m}$ to zero. The correspondence

can employ three mechanisms: (1) mass movement from a positive region to a negative region, with cost proportional to mass and distance, (2) mass destruction, with cost proportional to mass, and (3) mass creation, also with cost proportional to mass. The cost to create or destroy a unit of mass is $\gamma$.

Formally, we solve the following optimization:

$$\mathcal{T}(\boldsymbol{a}) = \min_{\boldsymbol{m},\boldsymbol{p},\boldsymbol{c}} \left[ \sum_{i,j} d_{ij} m_{ij} + \gamma \sum_i (p_i + c_i) \right] \tag{55}$$

subject to

$$a_i + p_i - c_i - \sum_j m_{ij} = 0 \quad \forall i \tag{56}$$

$$p_i > 0 \quad \forall i \tag{57}$$

$$c_i > 0 \quad \forall i \tag{58}$$

$$m_{ij} > 0 \quad \forall i,j \tag{59}$$

where $d_{ij}$ is the distance (Euclidean) from pixel $i$ to pixel $j$, $m_{ij}$ is the mass moved from pixel $i$ to pixel $j$, and $p_i$ and $c_i$ are the mass production and consumption at pixel $i$. Both $i$ and $j$ are in the range $1 \ldots h \times w$.

At first glance, this problem may seem computationally infeasible because the number of $d_{ij}$ parameters equals the number of pixels squared, e.g., a trillion parameters for a one-megapixel image. Fortunately, we can reduce the number of parameters to $\sim (h \times w)$ by pruning all edges where $d_{ij} > 2\gamma$. Intuitively, if $d_{ij} > 2\gamma$, the cost to destroy a unit of mass at location $i$ and recreate it at location $j$ is less than the cost to move it from $i$ to $j$.

Despite pruning, this optimization remains somewhat impractical. Finding a solution takes $\sim$seconds using the open-source solvers available in SciPy. These runtimes make loss evaluation the primary bottleneck during training. We might be able to reduce runtime using other solvers—either general-purpose commercial solvers or specialized optimal transport solvers. We leave this as future work.

## C  Composing Stabilizers

Often, a deployed model will be faced with several simultaneous corruptions. One option in this scenario is to train a single stabilizer on the expected combination of corruptions. However, doing so requires full knowledge of the corruptions at training time. In this section, we explore an alternate approach: composing (fusing) single-purpose stabilizers without additional training.

We consider controlled stabilizers without spatial fusion. Assume we have two single-corruption stabilizers, denoted by superscripts 1 and 2. The output $\tilde{z}_{i,t}$ of the fused stabilizers at layer $i$ is

$$\tilde{z}_{i,t} = \boldsymbol{\beta}_{i,t}^2 \odot \tilde{z}_{i,t}^1 + (1 - \boldsymbol{\beta}_{i,t}^2) \odot \tilde{z}_{i,t-1}, \tag{60}$$

$$\tilde{z}_{i,t}^1 = \boldsymbol{\beta}_{i,t}^1 \odot z_{i,t} + (1 - \boldsymbol{\beta}_{i,t}^1) \odot \tilde{z}_{i,t-1}^1, \tag{61}$$

$$\boldsymbol{\beta}_{i,t}^1 = \sigma(h_i^1(g^1(\boldsymbol{x}_t, \boldsymbol{x}_{t-1}))), \tag{62}$$

$$\boldsymbol{\beta}_{i,t}^2 = \sigma(h_i^2(g^2(\boldsymbol{x}_t, \boldsymbol{x}_{t-1}))). \tag{63}$$

Note that we have removed the feature-space inputs $z_t$, $\tilde{z}_{t-1}$, and $z_{t-1}$ to the stabilization heads. In experiments, we found this was necessary to prevent unintended interactions between the stabilizers.

The above formulation is roughly equivalent to applying a single stabilizer with decay $\boldsymbol{\beta}_{i,t}^1 \odot \boldsymbol{\beta}_{i,t}^2$. We can modify the method slightly to make this strictly true, thereby obtaining a commutative composition. We replace $\tilde{z}_{i,t-1}^1$ in Equation 61 with $\tilde{z}_{i,t-1}$, obtaining

$$\tilde{z}_{i,t} = \boldsymbol{\beta}_{i,t}^2 \odot (\boldsymbol{\beta}_{i,t}^1 \odot z_{i,t} + (1 - \boldsymbol{\beta}_{i,t}^1) \odot \tilde{z}_{i,t-1}) + (1 - \boldsymbol{\beta}_{i,t}^2) \odot \tilde{z}_{i,t-1}, \tag{64}$$

$$= \boldsymbol{\beta}_{i,t}^2 \odot \boldsymbol{\beta}_{i,t}^1 \odot z_{i,t} + (\boldsymbol{\beta}_{i,t}^2 - \boldsymbol{\beta}_{i,t}^2 \odot \boldsymbol{\beta}_{i,t}^1 + 1 - \boldsymbol{\beta}_{i,t}^2) \odot \tilde{z}_{i,t-1}, \tag{65}$$

$$= \boldsymbol{\beta}_{i,t}^2 \odot \boldsymbol{\beta}_{i,t}^1 \odot z_{i,t} + (1 - \boldsymbol{\beta}_{i,t}^2 \odot \boldsymbol{\beta}_{i,t}^1) \odot \tilde{z}_{i,t-1}. \tag{66}$$

Intuitively, the fused stabilizer retains the current features $z_{t,i}$ only if both decays are near one; this indicates that neither controller backbone detected corruption-induced instability.

In our experiments, we use the initial non-commutative version (it was slightly easier to implement). However, we expect the two formulations to give similar results in general.

# D   Method Details

This section provides further details of our method and implementation, including the particular architecture we use for the controller, our approach for training initialization, and a more formal definition of the spatial fusion mechanism.

## D.1   Controller Architecture

The controller backbone uses a simple convolutional architecture with 7 layers. The backbone has 32 channels for HDRNet and NAFNet, and 16 channels for Depth Anything. Controller heads have 4 layers. The number of channels in the last head layer depends on the shape of $z$ and the size of the fusion kernel. For other head layers, we use 64 channels for HDRNet and NAFNet, and 32 channels for Depth Anything. All convolutions use a $3 \times 3$ kernel, and all except the head outputs are followed by a leaky ReLU with negative slope 0.01.

## D.2   Initialization

When training, we initialize such that the predicted $\beta$ values are near 1. This corresponds to a rapid decay of the past state and less stabilization; i.e., the stabilizers are initialized close to an identity. For controlled stabilizers, this can be achieved by adding a final bias to $h$ (before the sigmoid) and initializing this bias to a sufficiently positive value $v$. For the simple learned stabilizer (without a controller), the trained parameters are logit values $l$ used to generate a decay $\in [0, 1]$ via $\beta = \sigma(l)$. Again, we initialize these logits to a positive value $v$. For both the simple learned and controlled variants, we find that $v = 4$ (resulting in $\beta \approx 0.98$) works well.

## D.3   Spatial Fusion

Let $\mathcal{N}$ be a spatial neighborhood around the pixel to be stabilized. Let $c$ denote the channel index, $j$ the index of the stabilized pixel, $k \in \mathcal{N}$ the neighbor pixel index, and $l \in 1 : m$ an index into the kernel $\boldsymbol{\eta}$. The output $\tilde{z}_{t,c,j}$ of the spatial fusion stabilizer is given by

$$\tilde{z}_{t,c,k} = \eta_{t,c,m+1} z_{t,c,j} + \sum_{(k,l) \in (\mathcal{N}, 1:m)} \eta_{t,c,l} \tilde{z}_{t,c,k} \tag{67}$$

$$\boldsymbol{\eta}_{t,c} = \text{Softmax}([h(g(\boldsymbol{x}_t, \boldsymbol{x}_{t-1}), \boldsymbol{z}_t, \boldsymbol{z}_t - 1)]_{cm:cm+m-1}, 0), \tag{68}$$

The summation iterates over locations in the neighborhood, with the kernel index $l$ always corresponding to the neighbor index $k$. The indexing on the output of $h$ extracts the $m$ channels needed to construct the kernel $\boldsymbol{\eta}_{t,c}$. Note that we have dropped the layer index $i$ here for brevity.

# E   Experiment Details

This section contains experiment details and hyperparameter values.

## E.1   Image Enhancement

**Base model fine-tuning.** We train for 80 epochs (2k iterations per epoch), using the Adam optimizer [31], an MSE loss, and batches of 8 randomly sampled frames. The learning rate is initially set to $10^{-4}$ and is scaled by 0.1 after epochs 40 and 60.

**Stabilizer training and evaluation.** Each training batch consists of one randomly sampled video snippet containing $\tau = 8$ consecutive frames. Gradients are computed using BPTT. Stabilizers are trained for 20 epochs (4k iterations per epoch) using the Adam optimizer and our unified loss with $\delta = \|\cdot\|_2$. The learning rate is initialized to $10^{-3}$ for the simple learned stabilizer and $10^{-4}$ for other variants, and is reduced by a factor of 10 after epochs 10 and 15. We evaluate stabilizers on the validation set, processing each video in a single pass per video (i.e., $\tau \gg 8$ at evaluation).

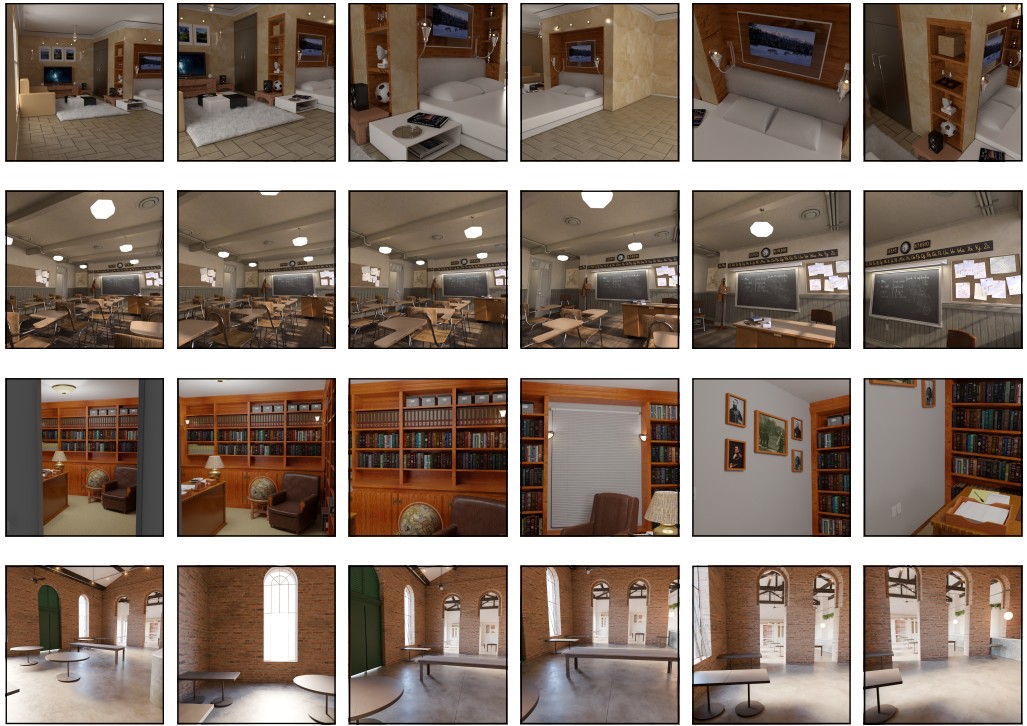

Figure 6: **VisionSim sequences.** Showing scenes bachelors-quarters, classroom, library-homeoffice, and restaurant. For each, we show frames 10, 110, 210, 310, 410, and 510.

## E.2 Video Denoising

**Base model fine-tuning.** Each batch consists of 8 randomly sampled, randomly cropped patches of size 256×256. We train for 20 epochs (2k steps per epoch) with Adam and an MSE loss. We set the initial learning rate to $10^{-4}$, scaling by 0.1 after epochs 10 and 15.

**Stabilizer training and evaluation.** Each batch consists of one randomly sampled, randomly cropped video snippet of size 256×256 containing $\tau = 8$ consecutive frames. We train for 20 epochs (2k steps per epoch) using Adam and the unified loss with $\delta = || \cdot ||_2$. The initial learning rate is set to $10^{-2}$ for the simple learned stabilizer and $10^{-4}$ for the controlled and spatial variants. In all cases, we scale the learning rate by 0.1 after epochs 10 and 15. We evaluate stabilizers on the validation set in a single pass per video ($\tau \gg 8$).

## E.3 Corruption Robustness

**Depth training dataset.** We use the VisionSim [25] framework to generate a dataset with ground-truth depth labels, according to the instructions provided at the following URL:
`https://visionsim.readthedocs.io/en/latest/tutorials/large-dataset.html`
The resulting dataset contains 50 scenes with 59950 total frames and resolution 800×800. All scenes are indoor and exclusively contain ego-motion. See Figure 6 for several representative sequences.

**Depth metrics.** Depth Anything predicts relative disparity (inverse depth), which requires an affine alignment to the ground truth before computing metrics (see [54] for details). After this alignment, we evaluate the standard AbsRel and Delta–1 ($\delta > 1.25$) metrics. We exclude outliers by clipping the aligned depth to a maximum of 200. It is critical to measure instability *after* alignment; otherwise, the network can achieve arbitrarily low instability without harming AbsRel and Delta–1 by scaling predictions to a small range around zero.

**Depth stabilizer training.** We train Depth Anything stabilizers on randomly sampled snippets of length $\tau = 8$, randomly cropped to size 512×512. We train using Adam for 20 epochs (4k iterations

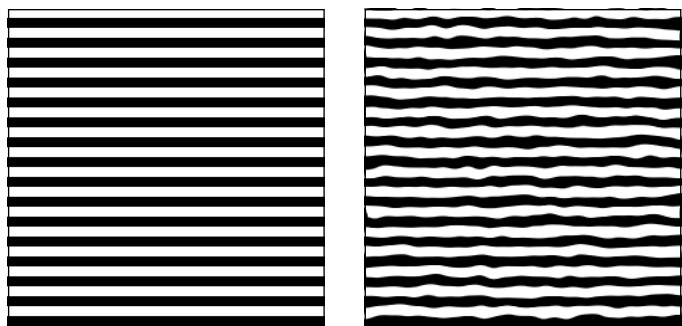

Figure 7: **Elastic transform.** A dummy image before and after applying the elastic transform, using the same parameters as our experiments (magnitude $\alpha = 50.0$ and smoothness $\sigma = 5.0$). The image size is 256×256, and the lines have width 16.

per epoch) with a batch size of one. The learning rate is initialized to $10^{-4}$ and reduced by 0.1 after epochs 10 and 15. We evaluate the unified loss ($\delta = || \cdot ||_2$) after affine alignment.

**Image corruptions.** We generate elastic distortions using the Torchvision `ElasticTransform` class with default settings (magnitude $\alpha = 50.0$ and smoothness $\sigma = 5.0$). See Figure 7 for a visualization of the elastic transform.

For image denoising, we add all corruptions *after* Gaussian noise. For example, dropped frames contain zeros, not zero-mean Gaussian noise.

### E.4 Adverse Weather Robustness

**Base model fine-tuning.** We use the same schedule and hyperparameters as in Section E.2. Each epoch consists of 1k training steps.

**Stabilizer training and evaluation.** We again use the same schedule and hyperparameters as in Section E.2. The initial learning rate is set to $10^{-4}$, and each epoch consists of 800 steps.

### E.5 Compute Requirements

We train and evaluate on a compute cluster largely using RTX A4500 GPUs. Fine-tuning and stabilizer training take 1–2 days on a single GPU, and full evaluation takes 1–3 hours.

## F Additional Results

### F.1 Stabilizer Composition

We evaluate composed stabilizers (Appendix C) on the NFS denoising task ($\sigma = 0.1$) using the NAFNet model. We train single-purpose corruption stabilizers using the same hyperparameters as other experiments, then evaluate composition under all possible two-corruption pairings. Results are shown in Table 3. Generally, stabilizer composition is effective if the second corruption does not significantly change the appearance of the first. For example, JPEG → impulse works better than impulse → JPEG. JPEG compression obscures high-frequency impulse noise, thereby interfering with the impulse noise controller backbone. This limitation could likely be addressed through data augmentation during stabilizer training, which would make the backbones more robust to changes in corruption appearance.

### F.2 Semantic Segmentation

**Task, dataset, and base model.** In this subsection, we analyze a higher-level prediction task: semantic segmentation. We use the DeepLabv3+ model [8]—specifically, the MobileNet variant published by [15]. We train and evaluate on the VIPER dataset [56], which contains video sequences captured in a game engine (GTA V) and automatically labeled for various vision tasks. The predefined

| First corruption | Second corruption | Ours (stabilized) | | | Base model (unstabilized) | | |
|---|---|---|---|---|---|---|---|
| | | PSNR | SSIM | Instability | PSNR | SSIM | Instability |
| Patch drop | Elastic distortion | 29.53 | 0.838 | 22.64 | 18.56 | 0.660 | 152.40 |
| Patch drop | Frame drop | 33.88 | 0.919 | 19.15 | 17.47 | 0.658 | 218.69 |
| Patch drop | JPEG artifacts | 29.07 | 0.845 | 31.67 | 18.51 | 0.637 | 152.60 |
| Patch drop | Impulse noise | 24.83 | 0.769 | 60.89 | 17.49 | 0.367 | 153.96 |
| Elastic distortion | Patch drop | 29.93 | 0.850 | 19.24 | 18.42 | 0.657 | 155.24 |
| Elastic distortion | Frame drop | 28.01 | 0.780 | 17.79 | 25.65 | 0.769 | 129.61 |
| Elastic distortion | JPEG artifacts | 29.11 | 0.842 | 18.46 | 25.94 | 0.759 | 53.75 |
| Elastic distortion | Impulse noise | 27.20 | 0.785 | 16.22 | 22.95 | 0.480 | 69.70 |
| Frame drop | Patch drop | 33.84 | 0.916 | 19.79 | 17.50 | 0.659 | 218.18 |
| Frame drop | Elastic distortion | 29.67 | 0.844 | 18.69 | 25.65 | 0.768 | 130.22 |
| Frame drop | JPEG artifacts | 25.99 | 0.737 | 17.59 | 26.56 | 0.720 | 124.21 |
| Frame drop | Impulse noise | 28.95 | 0.807 | 57.28 | 23.14 | 0.467 | 103.00 |
| JPEG artifacts | Patch drop | 30.55 | 0.857 | 21.88 | 18.41 | 0.638 | 154.20 |
| JPEG artifacts | Elastic distortion | 29.19 | 0.844 | 18.16 | 25.81 | 0.755 | 54.48 |
| JPEG artifacts | Frame drop | 25.87 | 0.737 | 17.16 | 26.56 | 0.720 | 124.21 |
| JPEG artifacts | Impulse noise | 28.40 | 0.810 | 19.74 | 23.93 | 0.475 | 65.24 |
| Impulse noise | Patch drop | 30.14 | 0.831 | 24.61 | 16.98 | 0.351 | 163.34 |
| Impulse noise | Elastic distortion | 26.73 | 0.790 | 17.86 | 23.38 | 0.600 | 62.01 |
| Impulse noise | Frame drop | 28.94 | 0.794 | 21.21 | 22.66 | 0.460 | 138.58 |
| Impulse noise | JPEG artifacts | 25.25 | 0.599 | 39.47 | 21.55 | 0.352 | 89.25 |

Table 3: **Stabilizer composition.** The effectiveness of composing stabilizers for different combinations and orderings of input corruptions. Results are for NAFNet on the NFS dataset with moderate noise ($\sigma = 0.1$). The base model evaluated without corruptions achieves PSNR 36.6, SSIM 0.945, and instability 26.2.

training and validation splits contain 77 sequences (134097 frames) and 47 sequences (49815 frames), respectively. We measure prediction quality using pixel accuracy and mIoU on the predefined validation set. Due to the discrete nature of predictions, we report categorical instability—the fraction of pixels whose category changes between frames (equivalent to $|| \cdot ||_0$).

**Experiment protocol.** We fine-tune the unstabilized model, starting with the Cityscapes [12] weights published by [15]. A $1\times1$ convolution is applied to the output logits to adapt the number of classes for VIPER. We fine-tune for 60 epochs (1925 batches per epoch), using a batch size of 16 and a cross-entropy loss. Adam was used with an initial learning rate of $10^{-4}$, scaled by 0.1 after epochs 20 and 40.

After fine-tuning, stabilizers are attached to the model input, the model output, each InvertedResidual layer, the `aspp` block, the `project` block, and the `classifier` block. We then freeze the fine-tuned weights and train stabilizers on snippets of length $\tau = 8$ with $\lambda = 0.4$. Here we tried both $|| \cdot ||_2$ and cross-entropy for $\delta$. Cross-entropy gave slightly better results, despite not satisfying the formal criteria in Section 4 (e.g., it is not symmetric). Therefore, we report results with cross-entropy in the remaining experiments.

When training stabilizers, each batch consists of one snippet of length $\tau = 8$ frames. We train for 60 epochs (3080 steps per epoch), using Adam with the same learning rate schedule as in fine-tuning.

**Results.** The unstabilized model achieves categorical instability 0.079, mIoU 0.406, and pixel accuracy 0.900. After adding stabilizers, we obtain instability 0.059, mIoU 0.411, and pixel accuracy 0.901. Similar to other tasks, there is a significant improvement in stability, along with an increase in accuracy (mIoU).

### F.3 Segmentation Robustness

**Task, dataset, and base model.** In this subsection, we evaluate the DeepLabv3+ segmentation model against the five corruptions from Section 6.3 (patch drop, elastic distortion, frame drop, JPEG artifacts, and impulse noise). We use the same dataset (VIPER) and metrics as in Section F.2.

**Experiment protocol.** We follow the same protocol as in Section F.2 when training and evaluating corruption stabilizers.

| Corruption | Method | mIoU | Accuracy | Instability |
|---|---|---|---|---|
| Patch drop | Base model | 0.060 | 0.164 | 0.454 |
| | Ours | 0.405 | 0.896 | 0.064 |
| Elastic distortion | Base model | 0.377 | 0.888 | 0.090 |
| | Ours | 0.403 | 0.898 | 0.064 |
| Frame drop | Base model | 0.369 | 0.829 | 0.209 |
| | Ours | 0.406 | 0.898 | 0.063 |
| JPEG artifacts | Base model | 0.109 | 0.313 | 0.216 |
| | Ours | 0.337 | 0.862 | 0.066 |
| Impulse noise | Base model | 0.051 | 0.300 | 0.312 |
| | Ours | 0.399 | 0.894 | 0.065 |

Table 4: **Segmentation robustness.** For all corruptions, our method simultaneously improves mIoU, pixel accuracy, and instability. The improvement is largest for corruptions that significantly change the appearance of the input, e.g., impulse noise.

**Results.** See Table F.3 for metric values, and Figure 13 for sample predictions. Our method improves both task metrics and stability across all corruptions. For patch drop, JPEG artifacts, and impulse noise, adding stabilizers allows the model to recover from catastrophic prediction failures.

## F.4   DAVIS Denoising

**Task, dataset, and base model.** In addition to NFS, we evaluate NAFNet denoising on the standard DAVIS benchmark [40, 52, 62]. DAVIS contains 50 videos (3455 frames) collected at 24 FPS. We use the dataset's predefined train/validation split and scale images to a short edge length of 480 [62]. Following from prior work [40, 62], we evaluate with a noise level of $40/255 \approx 0.16$.

**Experiment protocol.** We fine-tune the base model and train stabilizers following the same procedure as for NFS (see Sections 6.2 and E.2). Fine-tuning epochs contain 3k steps, and stabilizer training epochs contain 2.4k steps.

**Results.** See Figure 8 and Table 7 for results. Overall, the behavior is similar to the NFS results in Section 6.2. However, the "win-win" region (where both accuracy and stability are improved) is smaller for DAVIS. This is likely caused by DAVIS's $10\times$ lower frame rate, which corresponds to higher inter-frame motion and lower frame-to-frame correlation.

## F.5   Adversarial Robustness

In addition to natural corruptions, we evaluate our method in the presence of adversarial corruptions. We consider a setting where the attacker has knowledge of the base model and its parameters, but not of the stabilizers. We reason that because the stabilizers have fewer parameters than the base model and can be trained more quickly, a defender could update the stabilizer parameters after a weight leak rather than retrain the entire model.

**Task and model.** We evaluate adversarial robustness on a binary classification task derived from the DAVIS dataset. Frames are processed by tightly cropping around an object's segmentation mask, treating individual instances as separate images, and labeling each crop as human or nonhuman. As our backbone, we use ResNet-50 [22] pre-trained on ImageNet. We replace the original 1000-class output layer with a two-class linear layer.

**Experiment protocol.** We freeze all pre-trained weights except those in the final residual block (`layer4`) and the new classification head. We fine-tune these parameters for 100 epochs on the binary classification task. We start with a learning rate of $10^{-2}$, decreasing it by a factor of 0.1 at epochs 40 and 80, and employ frame-level data augmentation consisting of random rotation, horizontal flip, and color jitter. To preserve temporal context for future stabilizers, each training sample consists of a sequence of eight consecutive frames.

Then, we generate adversarial examples using the iterative Fast Gradient Sign Method (I-FGSM) with an overall perturbation bound $\varepsilon = 0.1$ and 20 iterations per image [19]. As the attacker does not know the ground-truth class, for each image, we randomly apply either the sign of the gradient step or its negation. We selected these hyperparameters because they produce a substantial drop in

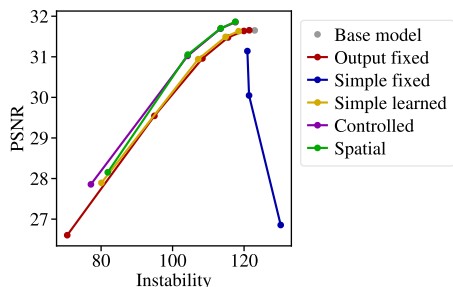

Figure 8: **DAVIS denoising.** We obtain the best quality/stability tradeoff when using a stabilization controller ("controlled" and "spatial"). On DAVIS, spatial fusion does not offer a significant advantage compared to a basic controlled stabilizer. DAVIS has relatively high inter-frame motion, which often exceeds the size of the spatial kernel (i.e., the maximum translation achievable with spatial fusion).

baseline accuracy, creating a clear opportunity for the stabilizers to improve performance. Then, to defend against the attack, we add controlled stabilizers to each bottleneck block of the ResNet model. All original ResNet parameters are frozen, and only stabilizers are trained for 20 epochs under the same I-FGSM attack settings as above. The stabilizer training uses an initial learning rate of $10^{-4}$, reduced by a factor of 0.1 after epochs 1 and 10, and each batch consists of two sequences of eight frames to ensure well-defined gradients. No additional augmentations are applied during this phase.

**Results.** Under our adversarial setup, the ResNet-50 fine-tuned baseline achieves 77.0% accuracy, while our stabilizer-augmented model reaches 88.8%, an absolute improvement of 11.8%. These results demonstrate that the proposed stabilizer modules can significantly improve resilience to adversarial attacks without requiring complete retraining.

### F.6 Spatial Fusion Failures

**Discussion.** In the denoising experiments (Section 6.2), we observed a significant PSNR reduction when using the spatial fusion method under extreme noise. This reduction only appears when evaluating on long sequences.

A closer examination of the outputs reveals blurring/ghosting artifacts that appear after some time has passed. These artifacts look like hard edges "bleeding out" into the surrounding regions. We believe these failures are related to the spatial fusion stabilizer on the output layer. Without this stabilizer, the network output is biased toward the current input frame (due to the network architecture, which predicts a noise residual). Adding a spatial fusion stabilizer to the output provides an independent mechanism for information to flow between pixels, thereby weakening this bias.

**Experiment protocol.** We ran an experiment to determine whether spatial fusion failures can be mitigated by training on longer sequences. We trained the spatial fusion stabilizer under extreme noise on sequences of length $\tau = 8$ (the default in our other experiments) and $\tau = 16$. We reduced the training patch size from $256 \times 256$ to $180 \times 180$ to compensate for increased training memory requirements on longer sequences. For $\tau = 8$, we doubled the number of iterations per epoch due to the lower number of frames in each iteration. We evaluated the resulting models on the full validation set containing long sequences.

**Results.** Training with $\tau = 8$ gave validation PSNR 22.70 and instability 11.04, whereas training with $\tau = 16$ gave PSNR 23.80 and instability 10.16. We expect this trend of improvement to hold as we further increase the training sequence length.

### F.7 Uncertainty Estimate

**Experiment protocol.** To estimate uncertainty in our results, we train and evaluate the spatial fusion stabilizer eight times with different random seeds. We consider image enhancement (HDRNet) for the moderate-strength local Laplacian operator ($\alpha = 0.25$). The random seed determines the stabilizer weight initialization and the training data shuffle. The training and evaluation protocol is identical to that in Sections 6.1 and E.1.

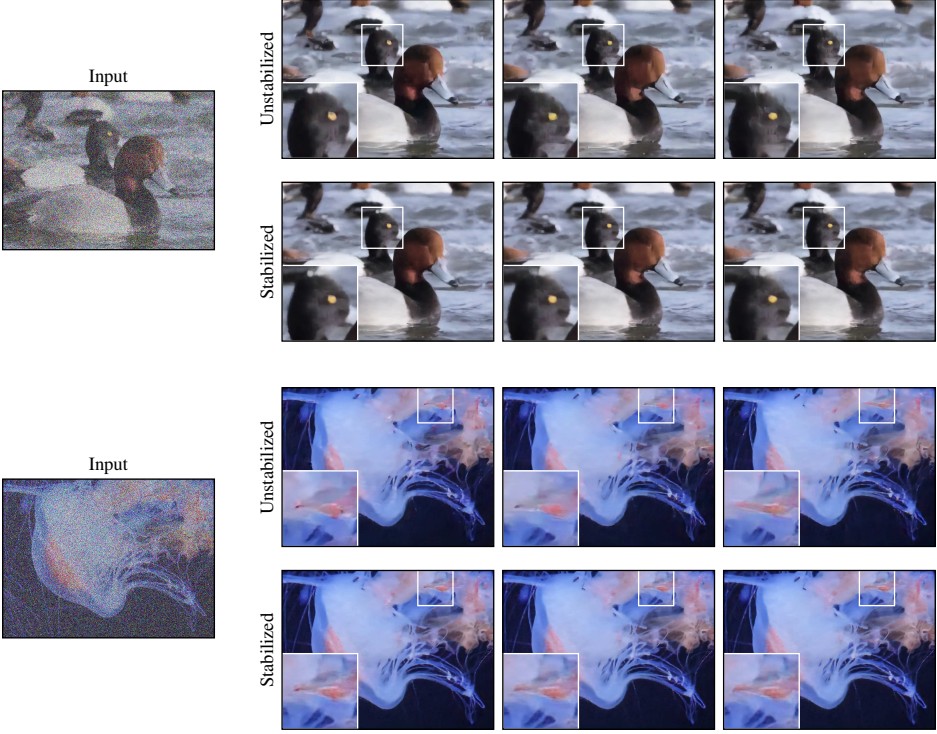

Figure 9: **Denoising under extreme noise.** As we increase the level of image noise, frame-wise temporal inconsistency becomes more severe, to the point that it becomes apparent when comparing static images. In the top sequence, we see shifting texture in the duck's head in the unstabilized features. In the jellyfish sequence, the denoiser hallucinates inconsistent spatial structure between frames. In both cases, adding a stabilizer noticeably improves temporal consistency. We encourage the reader to view the corresponding video files included with the supplement.

**Results.** PSNR ranges from a minimum of 32.17 to a maximum of 32.32, with a mean of 32.25. SSIM has range 0.926–0.929 (mean 0.927), and instability has range 28.61–28.80 (mean 28.71). For all metrics, variations around the mean are $< 0.5\%$.

### F.8 Figures

Figure 9 shows example sequences denoised under extreme Gaussian noise ($\sigma = 0.6$). We highlight regions of the image where instability is especially prominent. Differences are more noticeable in videos; we encourage the reader to view the video files included with the supplementary material.

Figures 10, 11, 12, and 13 contain examples of corruption robustness for image enhancement, denoising, depth estimation, and segmentation, respectively. Figure 14 illustrates improved weather robustness for denoising on RobustSpring 14.

### F.9 Tables

Table 5 provides complete results for image enhancement (Figure 4). Tables 6 and 7 contain complete results for denoising (Figures 5 and 8).

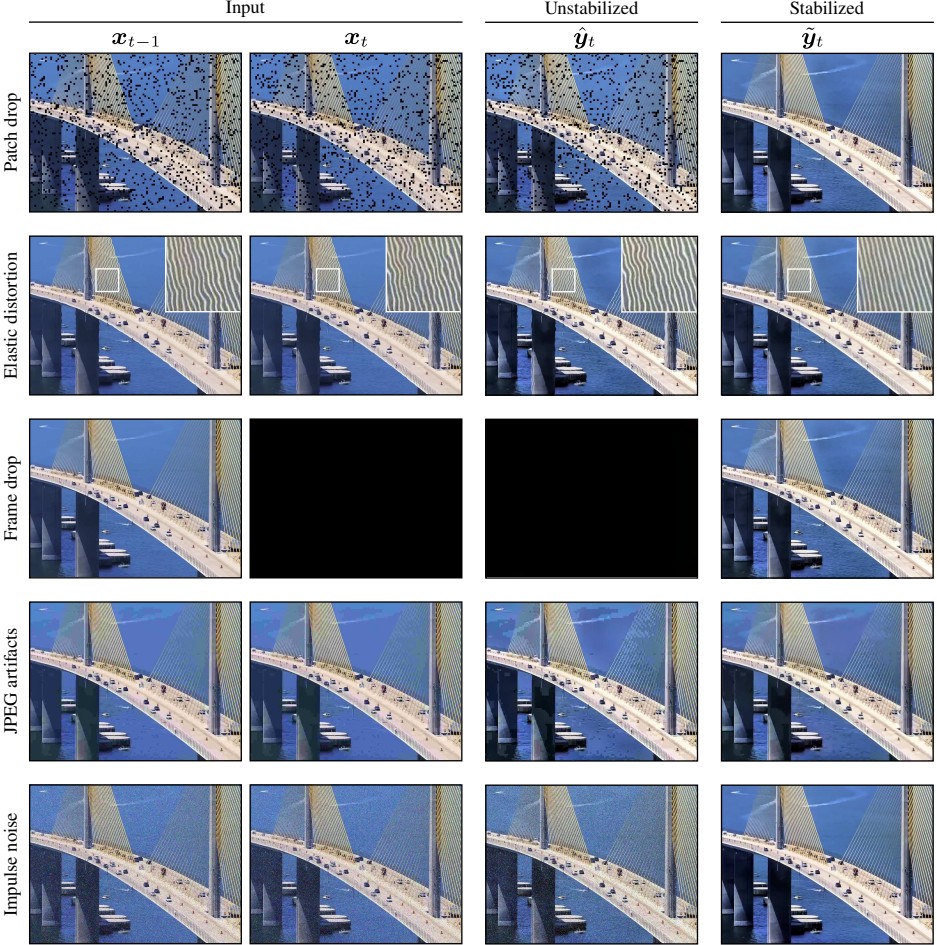

Figure 10: **Image enhancement robustness.** The effect of stabilization for image enhancement (HDRNet) under various image corruptions. See Section 6.3.

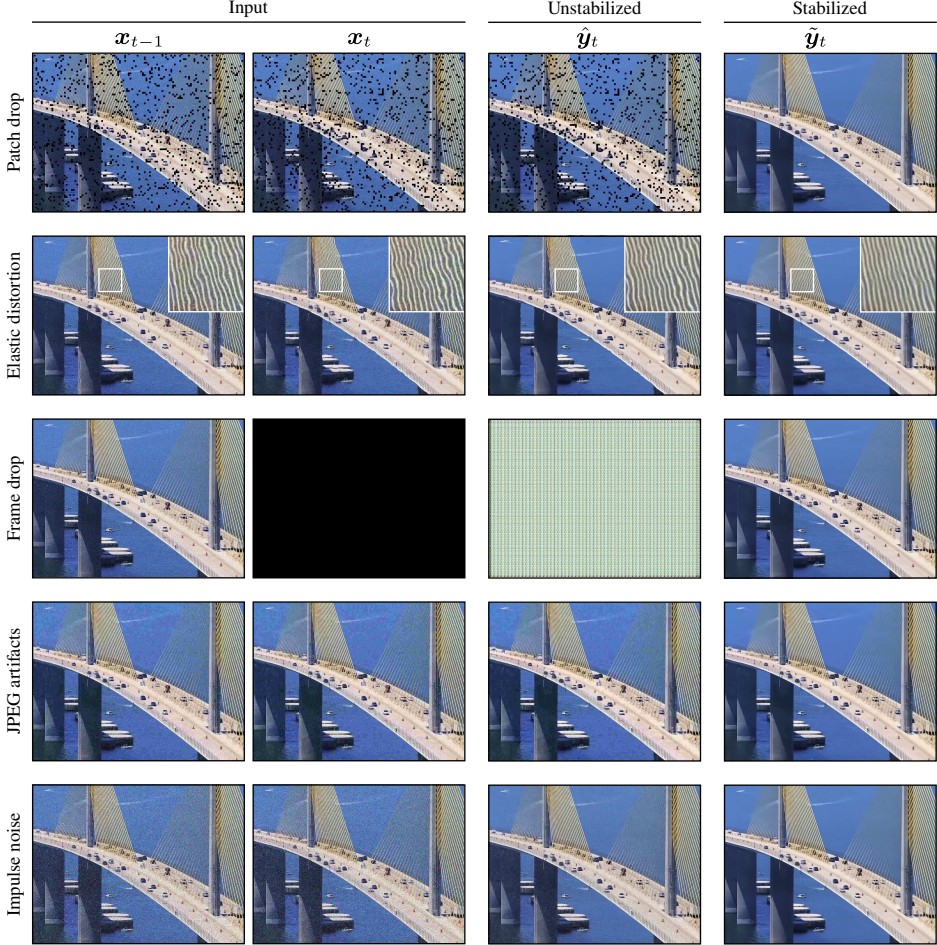

Figure 11: **Denoising robustness.** The effect of stabilization for denoising (NAFNet) under various image corruptions. See Section 6.3.

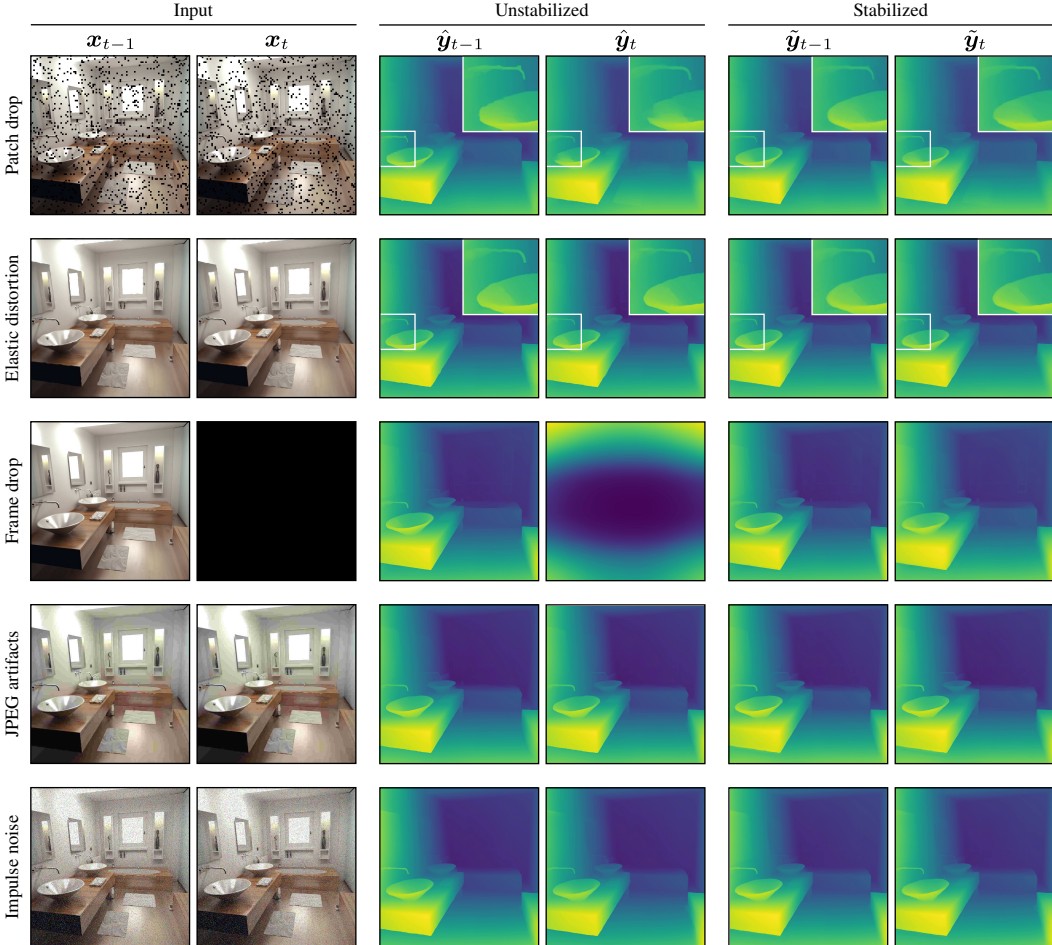

Figure 12: **Depth estimation robustness.** The effect of stabilization for depth estimation (Depth Anything v2) under various image corruptions. Improvements are most prominent for the patch drop, elastic distortion, and frame drop corruptions. The base model already has reasonable robustness to JPEG artifacts and noise. See Section 6.3.

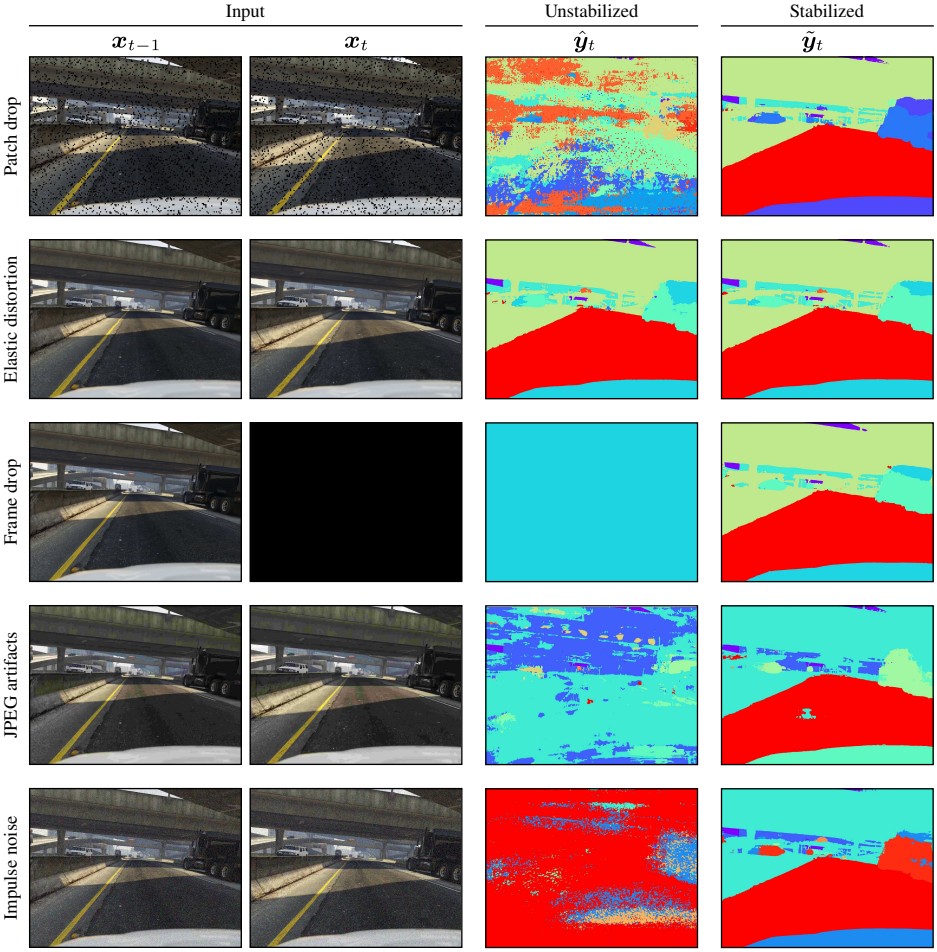

Figure 13: **Segmentation robustness.** The effect of stabilization for segmentation (DeepLabv3+) under various image corruptions. See Section F.2. Note that the method we use to generate the color map may cause color-class correspondences to vary across rows.

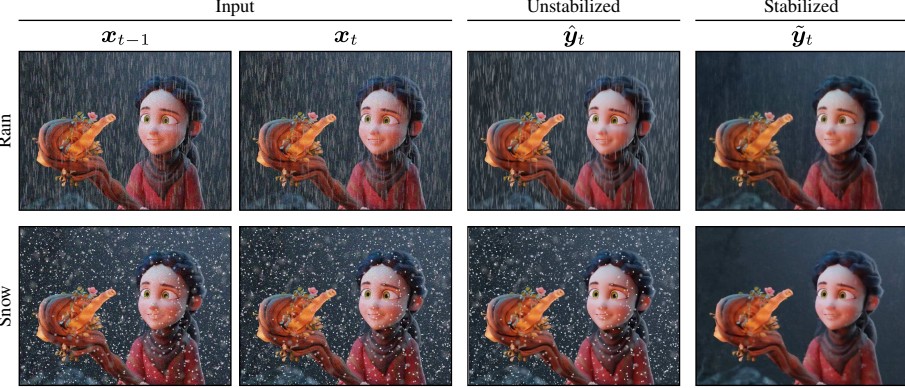

Figure 14: **Adverse weather robustness.** The effect of stabilization for denoising (NAFNet) under weather corruptions on the RobustSpring dataset. See Section 6.4.

| Method | Strength | Moderate intensity ($\alpha = 0.5$) | | | High intensity ($\alpha = 0.25$) | | |
| --- | --- | --- | --- | --- | --- | --- | --- |
| | | Instability | PSNR | SSIM | Instability | PSNR | SSIM |
| Gaussian | $\mu = 0.50$ | 29.46 | 30.70 | 0.917 | 36.26 | 25.73 | 0.852 |
| Gaussian | $\mu = 1.00$ | 22.23 | 29.26 | 0.894 | 27.20 | 24.90 | 0.824 |
| Gaussian | $\mu = 2.00$ | 16.49 | 27.38 | 0.848 | 19.99 | 23.65 | 0.773 |
| Gaussian | $\mu = 3.00$ | 13.34 | 26.18 | 0.811 | 16.09 | 22.79 | 0.732 |
| Gaussian | $\mu = 4.00$ | 11.29 | 25.33 | 0.782 | 13.57 | 22.15 | 0.700 |
| Gaussian | $\mu = 6.00$ | 8.76 | 24.18 | 0.739 | 10.49 | 21.26 | 0.654 |
| Output fixed | $\beta = 0.99$ | 32.74 | 30.92 | 0.919 | 40.38 | 25.84 | 0.856 |
| Output fixed | $\beta = 0.98$ | 32.42 | 30.91 | 0.919 | 39.99 | 25.83 | 0.856 |
| Output fixed | $\beta = 0.95$ | 31.50 | 30.87 | 0.919 | 38.82 | 25.81 | 0.855 |
| Output fixed | $\beta = 0.90$ | 30.00 | 30.76 | 0.917 | 36.94 | 25.76 | 0.852 |
| Output fixed | $\beta = 0.80$ | 27.16 | 30.40 | 0.912 | 33.37 | 25.57 | 0.846 |
| Output fixed | $\beta = 0.60$ | 21.80 | 29.26 | 0.893 | 26.65 | 24.91 | 0.824 |
| Simple fixed | $\beta = 0.99$ | 32.81 | 30.92 | 0.919 | 40.64 | 25.84 | 0.856 |
| Simple fixed | $\beta = 0.98$ | 32.55 | 30.90 | 0.919 | 40.47 | 25.83 | 0.855 |
| Simple fixed | $\beta = 0.95$ | 31.61 | 30.82 | 0.918 | 39.67 | 25.77 | 0.853 |
| Simple fixed | $\beta = 0.90$ | 29.76 | 30.59 | 0.914 | 37.59 | 25.62 | 0.848 |
| Simple fixed | $\beta = 0.80$ | 25.88 | 29.90 | 0.903 | 32.57 | 25.20 | 0.833 |
| Simple fixed | $\beta = 0.60$ | 19.28 | 28.10 | 0.865 | 23.85 | 23.99 | 0.788 |
| Simple learned | $\lambda = 0.1$ | 32.44 | 30.92 | 0.920 | 39.58 | 25.84 | 0.856 |
| Simple learned | $\lambda = 0.2$ | 31.34 | 30.87 | 0.919 | 37.64 | 25.79 | 0.854 |
| Simple learned | $\lambda = 0.4$ | 28.96 | 30.66 | 0.917 | 33.74 | 25.60 | 0.849 |
| Simple learned | $\lambda = 0.8$ | 23.60 | 29.75 | 0.903 | 26.43 | 24.90 | 0.825 |
| Controlled | $\lambda = 0.1$ | 31.78 | 31.25 | 0.922 | 38.31 | 26.26 | 0.865 |
| Controlled | $\lambda = 0.2$ | 30.57 | 31.13 | 0.920 | 36.11 | 26.11 | 0.859 |
| Controlled | $\lambda = 0.4$ | 28.00 | 30.86 | 0.917 | 31.98 | 25.92 | 0.855 |
| Controlled | $\lambda = 0.8$ | 21.79 | 29.85 | 0.901 | 24.64 | 25.17 | 0.824 |
| Spatial | $\lambda = 0.1$ | 32.38 | 32.81 | 0.934 | 40.32 | 27.97 | 0.891 |
| Spatial | $\lambda = 0.2$ | 31.29 | 32.70 | 0.933 | 38.29 | 27.79 | 0.890 |
| Spatial | $\lambda = 0.4$ | 28.66 | 32.30 | 0.928 | 34.22 | 27.45 | 0.881 |
| Spatial | $\lambda = 0.8$ | 22.31 | 31.00 | 0.910 | 25.61 | 26.27 | 0.849 |

Table 5: **Image enhancement results.** These results correspond to the experiments in Section 6.1 (Figure 4). We additionally include results for simple Gaussian smoothing of the output, where $\mu$ is the standard deviation of the smoothing kernel.

| Method | Strength | NFS moderate ($\sigma = 0.1$) | | | NFS strong ($\sigma = 0.2$) | | |
|---|---|---|---|---|---|---|---|
| | | Instability | PSNR | SSIM | Instability | PSNR | SSIM |
| Gaussian | $\mu = 0.50$ | 22.87 | 36.77 | 0.945 | 24.50 | 33.52 | 0.903 |
| Gaussian | $\mu = 1.00$ | 16.33 | 35.41 | 0.940 | 16.67 | 33.10 | 0.904 |
| Gaussian | $\mu = 2.00$ | 11.90 | 32.96 | 0.920 | 11.77 | 31.64 | 0.889 |
| Gaussian | $\mu = 3.00$ | 9.63 | 31.31 | 0.900 | 9.45 | 30.43 | 0.874 |
| Gaussian | $\mu = 4.00$ | 8.18 | 30.15 | 0.883 | 8.01 | 29.50 | 0.860 |
| Gaussian | $\mu = 6.00$ | 6.39 | 28.60 | 0.857 | 6.25 | 28.17 | 0.837 |
| Output fixed | $\beta = 0.99$ | 25.93 | 36.67 | 0.943 | 28.18 | 33.30 | 0.900 |
| Output fixed | $\beta = 0.98$ | 25.64 | 36.70 | 0.943 | 27.83 | 33.33 | 0.900 |
| Output fixed | $\beta = 0.95$ | 24.79 | 36.76 | 0.944 | 26.81 | 33.40 | 0.901 |
| Output fixed | $\beta = 0.90$ | 23.43 | 36.79 | 0.945 | 25.20 | 33.50 | 0.902 |
| Output fixed | $\beta = 0.80$ | 20.91 | 36.58 | 0.945 | 22.22 | 33.54 | 0.904 |
| Output fixed | $\beta = 0.60$ | 16.37 | 35.34 | 0.940 | 16.97 | 33.07 | 0.903 |
| Simple fixed | $\beta = 0.99$ | 27.40 | 35.95 | 0.927 | 32.19 | 32.22 | 0.847 |
| Simple fixed | $\beta = 0.98$ | 30.21 | 34.67 | 0.885 | 39.59 | 30.51 | 0.741 |
| Simple fixed | $\beta = 0.95$ | 40.44 | 31.30 | 0.734 | 63.31 | 26.52 | 0.491 |
| Simple fixed | $\beta = 0.90$ | 53.94 | 28.14 | 0.565 | 91.84 | 23.11 | 0.322 |
| Simple fixed | $\beta = 0.80$ | 65.09 | 25.62 | 0.437 | 114.95 | 20.51 | 0.226 |
| Simple fixed | $\beta = 0.60$ | 56.70 | 24.89 | 0.405 | 100.92 | 19.83 | 0.205 |
| Simple learned | $\lambda = 0.1$ | 22.98 | 36.82 | 0.947 | 22.34 | 33.58 | 0.908 |
| Simple learned | $\lambda = 0.2$ | 21.80 | 36.74 | 0.947 | 20.58 | 33.51 | 0.908 |
| Simple learned | $\lambda = 0.4$ | 19.46 | 36.36 | 0.946 | 17.63 | 33.21 | 0.907 |
| Simple learned | $\lambda = 0.8$ | 15.38 | 35.00 | 0.939 | 13.48 | 32.24 | 0.898 |
| Controlled | $\lambda = 0.1$ | 22.21 | 37.51 | 0.952 | 21.41 | 34.36 | 0.916 |
| Controlled | $\lambda = 0.2$ | 21.28 | 37.41 | 0.951 | 20.21 | 34.30 | 0.915 |
| Controlled | $\lambda = 0.4$ | 19.01 | 37.00 | 0.949 | 17.24 | 33.93 | 0.912 |
| Controlled | $\lambda = 0.8$ | 14.61 | 35.76 | 0.942 | 12.95 | 33.05 | 0.903 |
| Spatial | $\lambda = 0.1$ | 22.13 | 37.65 | 0.953 | 21.08 | 34.52 | 0.917 |
| Spatial | $\lambda = 0.2$ | 21.10 | 37.57 | 0.952 | 19.80 | 34.45 | 0.917 |
| Spatial | $\lambda = 0.4$ | 18.94 | 37.15 | 0.950 | 16.99 | 34.06 | 0.913 |
| Spatial | $\lambda = 0.8$ | 14.25 | 34.94 | 0.931 | 11.84 | 32.53 | 0.895 |

Table 6: **Denoising results, part 1/2.** These results correspond to the experiments in Section 6.2 (Figure 5). We additionally include SSIM and results for simple Gaussian smoothing of the output, where $\mu$ is the standard deviation of the smoothing kernel. See Table 7 for the second half of the data.

| Method | Strength | NFS extreme ($\sigma = 0.6$) | | | DAVIS ($\sigma = 40/255$) | | |
|---|---|---|---|---|---|---|---|
| | | Instability | PSNR | SSIM | Instability | PSNR | SSIM |
| Gaussian | $\mu = 0.50$ | 30.23 | 27.98 | 0.794 | 105.62 | 30.67 | 0.864 |
| Gaussian | $\mu = 1.00$ | 18.55 | 28.26 | 0.804 | 70.68 | 26.18 | 0.795 |
| Gaussian | $\mu = 2.00$ | 11.82 | 28.02 | 0.803 | 45.38 | 23.22 | 0.717 |
| Gaussian | $\mu = 3.00$ | 9.12 | 27.59 | 0.797 | 34.27 | 21.92 | 0.677 |
| Gaussian | $\mu = 4.00$ | 7.59 | 27.17 | 0.790 | 27.95 | 21.15 | 0.653 |
| Gaussian | $\mu = 6.00$ | 5.84 | 26.45 | 0.778 | 21.01 | 20.25 | 0.626 |
| Output fixed | $\beta = 0.99$ | 35.64 | 27.72 | 0.787 | 121.46 | 31.65 | 0.873 |
| Output fixed | $\beta = 0.98$ | 35.13 | 27.75 | 0.788 | 119.95 | 31.64 | 0.872 |
| Output fixed | $\beta = 0.95$ | 33.65 | 27.83 | 0.790 | 115.50 | 31.48 | 0.871 |
| Output fixed | $\beta = 0.90$ | 31.30 | 27.94 | 0.793 | 108.36 | 30.96 | 0.867 |
| Output fixed | $\beta = 0.80$ | 26.98 | 28.12 | 0.798 | 94.92 | 29.54 | 0.853 |
| Output fixed | $\beta = 0.60$ | 19.53 | 28.27 | 0.804 | 70.52 | 26.60 | 0.810 |
| Simple fixed | $\beta = 0.99$ | 45.83 | 26.35 | 0.616 | 120.92 | 31.14 | 0.844 |
| Simple fixed | $\beta = 0.98$ | 64.92 | 24.36 | 0.415 | 121.40 | 30.05 | 0.777 |
| Simple fixed | $\beta = 0.95$ | 119.74 | 20.09 | 0.191 | 130.25 | 26.85 | 0.581 |
| Simple fixed | $\beta = 0.90$ | 180.81 | 16.56 | 0.105 | 148.51 | 23.76 | 0.415 |
| Simple fixed | $\beta = 0.80$ | 230.73 | 13.86 | 0.067 | 164.16 | 21.26 | 0.309 |
| Simple fixed | $\beta = 0.60$ | 205.99 | 12.97 | 0.059 | 137.79 | 20.09 | 0.270 |
| Simple learned | $\lambda = 0.1$ | 20.24 | 28.29 | 0.812 | 118.48 | 31.63 | 0.873 |
| Simple learned | $\lambda = 0.2$ | 17.73 | 28.27 | 0.813 | 114.84 | 31.49 | 0.871 |
| Simple learned | $\lambda = 0.4$ | 14.25 | 28.13 | 0.812 | 107.17 | 30.94 | 0.865 |
| Simple learned | $\lambda = 0.8$ | 10.21 | 27.69 | 0.806 | 80.09 | 27.89 | 0.821 |
| Controlled | $\lambda = 0.1$ | 19.47 | 29.22 | 0.824 | 117.58 | 31.85 | 0.876 |
| Controlled | $\lambda = 0.2$ | 17.26 | 29.15 | 0.824 | 113.44 | 31.69 | 0.873 |
| Controlled | $\lambda = 0.4$ | 13.67 | 28.92 | 0.822 | 104.21 | 31.02 | 0.864 |
| Controlled | $\lambda = 0.8$ | 10.12 | 28.45 | 0.813 | 77.15 | 27.86 | 0.815 |
| Spatial | $\lambda = 0.1$ | 15.20 | 22.00 | 0.665 | 117.57 | 31.86 | 0.876 |
| Spatial | $\lambda = 0.2$ | 16.55 | 21.73 | 0.637 | 113.49 | 31.70 | 0.874 |
| Spatial | $\lambda = 0.4$ | 9.09 | 22.28 | 0.692 | 104.26 | 31.05 | 0.865 |
| Spatial | $\lambda = 0.8$ | 11.66 | 22.45 | 0.669 | 81.86 | 28.15 | 0.824 |

Table 7: **Denoising results, part 2/2.** These results correspond to the experiments in Section 6.2 (Figure 5). We additionally include SSIM and results for simple Gaussian smoothing of the output, where $\mu$ is the standard deviation of the smoothing kernel. See Table 6 for the first half of the data.

# G   Licenses and Copyright

**Code.** We use our own implementation of HDRNet. For NAFNet, we use the authors' code (`https://github.com/megvii-research/NAFNet`, MIT license). For Depth Anything, we use the `depth-anything/Depth-Anything-V2-Small-hf` HuggingFace module (available under an Apache 2.0 license). VisionSim is released under the MIT license. All scenes are licensed under a Creative Commons variant; see this Google Drive folder for attributions and further license details: `https://drive.google.com/drive/folders/1gRxhL3rbGDTfgKytre8WkbBu-QDJFy15`

**Datasets and assets.** We were unable to find license information for the Need for Speed dataset. DAVIS uses the BSD license. We use frames from the following YouTube videos in our figures:

- `https://www.youtube.com/watch?v=ANeMCOpx_84`
- `https://www.youtube.com/watch?v=HZ8VF0EdITk`
- `https://www.youtube.com/watch?v=MPZb9EQ3Wjs`
- `https://www.youtube.com/watch?v=obSH5F2DYvk`

