# OpenReview forum: "Instant Video Models: Universal Adapters for Stabilizing Image-Based Networks"
_NeurIPS.cc/2025/Conference — NeurIPS 2025 poster_

### Official Review · Reviewer_cmJ7 · 2025-06-16

**Clarity:** 3
**Significance:** 3
**Originality:** 3
**Rating:** 4
**Confidence:** 2

**Summary:**

This paper addresses the challenge of temporal inconsistency and corruption sensitivity in frame-based neural networks for video inference. The authors introduce a general approach using stabilization adapters that can be inserted into different architectures, combined with a resource-efficient training process that keeps the base network frozen. They propose a unified accuracy-stability-robustness loss to balance output fidelity, temporal consistency, and robustness, supported by theoretical analysis identifying conditions to avoid over-smoothing. The adapters operate causally on both internal features and outputs, generating control signals from spatiotemporal context to modulate predictions. Experiments on different tasks demonstrate the effectiveness of the proposed method.

**Questions:**

Please see the weaknesses part.

**Ethical Concerns:**

["NO or VERY MINOR ethics concerns only"]

**Final Justification:**

Thanks for the authors' reply. I have no further questions.

**Limitations:**

yes

**Quality:**

3

**Strengths And Weaknesses:**

[Strengths]
1. The proposed method can be applied to different single-frame image inference architectures.
2. The strategy of freezing the base network and only training the adapter parameters during training significantly reduces computational overhead.
3. The method is supported by detailed theoretical analysis on accuracy-robustness trade-offs.

[Weaknesses]

Overall, the method proposed in this paper is good, but clarifying the following points would further enhance the work:

1. For different tasks discussed in the paper, it appears that only a single network architecture is used for testing. If different networks are switched (especially between different styles, such as convolution vs. Transformer, or complex vs. lightweight architectures), would the proposed method remain effective?
2. When the noise level is high (e.g., σ=0.6), the proposed method shows poor stability and is highly sensitive to the parameter λ, leading to difficulties in parameter tuning. Additionally, adjusting λ is not straightforward in most cases: when λ exceeds the collapse bound, it may cause prediction collapse, and the "stability-accuracy win-win region" for some tasks is narrow, requiring meticulous manual tuning.
3. The experimental tasks, including enhancement, denoising, and depth estimation, are all relatively mature fields. How does the proposed method perform on more challenging tasks, such as low-light enhancement and deblurring?

---

> ### Author Rebuttal · Authors · 2025-07-31
>
> We thank the reviewer for their feedback and respond to their comments below.
>
> **Generalization across architectures.** While we have not tried different architectures within a single task, we considered several architectures in our experiments: NAFNet for denoising, HDRNet for image enhancement, and Depth Anything v2 for depth estimation. Since the submission, we have also added results for segmentation using DeeplabV3; we will include these results in camera-ready. These architectures cover a range of designs and include both lightweight and heavyweight models. While we believe our results are sufficient to support the main claim of the paper regarding the generality of the proposed method, if the reviewer feels this is critical, we are open to evaluating another architecture for one of the existing tasks in the camera-ready version.
>
> **Tuning $\lambda$.** The failure mode under extreme noise ($\sigma = 0.6$) is not related to difficulty in tuning $\lambda$ but rather a combination of architecture- and task-specific factors. We have provided detailed analysis in response to the suggestions from reviewers LzQr and pzRZ.
>
> Overall we found tuning $\lambda$ to be straightforward, due in part to the guarantees provided by the oracle bound. In general, we found good results with $\lambda$ in the range 0.2-0.4. We did not find that the results were sensitive to fine adjustments of $\lambda$ - variations within the oracle bound give a smooth, well-behaved tradeoff between accuracy and stability.
>
> **Evaluation on other tasks.** We have added results on semantic segmentation since the submission and will include these in camera-ready. We would also point out that denoising and low-light enhancement are two sides of the same coin (although the noise model may need to be adjusted to, e.g., Poisson in low light).

---

> > ### Comment · Reviewer_cmJ7 · 2025-08-01
> >
> > Thanks for the authors' reply. I have no further questions.

---

### Official Review · Reviewer_yzUk · 2025-07-02

**Clarity:** 4
**Significance:** 4
**Originality:** 3
**Rating:** 5
**Confidence:** 3

**Summary:**

The submission presents a method to incorporate temporal consistency (stability) on deep neural networks designed for single-frame estimation, so that they can be applied on video data. The goal is being able to reutilize already expensively-trained networks so that they produce temporally-consistent predictions on video, as opposed to training dedicated networks for video. The method consists on designing smoothers (of the output or of intermediate feature layers) like moving averages and to learn the parameters to control the amount of smoothing. The method is tested on three tasks: image enhancement, image denoising and robustness to several types of input corruptions (JPEG artifacts, frame drop, etc.), and in comparison with several variations of the method and other baseline methods, producing, overall good results.

**Questions:**

- Biased by the duration tau? : Equations (1) to (5) seem to depend on the duration of the video, tau. Later, in the experiments, there seem to be some issues with short videos. Would the method produce different results if the formulas included a normalization factor (1/tau) , that is, divide by tau to avoid long videos to have a much larger influence on the stability and other quantities compared to shorter videos.

- Table 1; image denoising, Frame drop. Why is the proposed approach 5 dB below the base model? Please double check these numbers.

- Fig 2: it is not clear what the meaning of "tilts towards stabler predictions". I think I did not find described in the text how one can identify such zones on the diagrams.

**Ethical Concerns:**

["NO or VERY MINOR ethics concerns only"]

**Final Justification:**

The authors have addressed my comments in the reply. The explanations need to be included in the revised version.
I remain positive about this submission.

**Limitations:**

yes

**Paper Formatting Concerns:**

Overall, the layout is correct. The authors did not try to reduce the the spacing to squeeze more content on the main manuscript.

**Quality:**

3

**Strengths And Weaknesses:**

Strengths:
- The submission is very clearly written; one can grasp the main ideas on the first pass.
- The motivation is well justified, given that large models (like Depth Anything) can only be trained by large companies with resources. It makes a lot of sense to be able to reutilize them and add some "controller" on top of them to leverage such models.
- The approach is sound, combining controllers, stabilizers and some theoretical analysis.
- Experiments are comprehensive, on three tasks and producing good results in comparison with several variations of the methods as well as other baselines.
- The results in the supplementary video are impressive.

Weaknesses:
- The spatial fusion seems to produce good results in most cases, but it is only shortly introduced at the end of Sec. 5. Please consider explaining it in more detail in a revised version.
- Figures 4 & 5: consider adding an arrow of which part of the space is the desired region. I assume it is the top left because it would be high PSNR and stable, but it is not clear if this is the goal. Please also consider adding curves to join points of the same type, so that we could see these curves, almost like approaching the Pareto front.

---

> ### Author Rebuttal · Authors · 2025-07-31
>
> We thank the reviewer for their feedback and respond to their comments below.
>
> **Explanation of spatial fusion.** We have included additional details in Section B.3 of the supplementary material, although we agree that the overall clarity of these explanations could be improved. We will revise and expand our description of spatial fusion in the camera-ready version.
>
> **Improvements to plots.** We agree that these changes would improve the readability of the plots, and will incorporate them in camera-ready.
>
> **Potential bias by duration $\tau$.** In our current experiments, we fix $\tau$ to a constant value during training, so there is no risk of longer sequences dominating the loss. However, for training setups where the video duration varies widely between sequences, it would be a good idea to introduce a duration-balancing factor $1 / \tau$ as suggested. We will add a note to Section 3 describing this consideration.
>
> **Reduction in PSNR.** We have double-checked these numbers and this does not appear to be a transcription error or a bug. We have also confirmed that our method gives considerably higher subjective quality on test videos (as can be seen in the supplement video).
>
> We conjecture that there are two things going on here. First, PSNR (averaged over frames in this case) does not capture the jarring perceptual effect of the dropped frames - so the high PSNR of the base model is somewhat deceptive. This points to the well-known limitations of PSNR as a video quality metric. We would point to the significant improvement in temporal stability as quantitative evidence of improved perceptual quality.
>
> Second, there are hints of the spatial-fusion denoising failure mode here - PSNR is much higher (exceeding the baseline) on shorter sequences. See the response to reviewers LzQr and pzRZ for an explanation and a potential solution.
>
> **Figure 2 explanation.** By "tilt toward" here we mean "slopes more gently toward." That is, deviations from the correct prediction are penalized less heavily if they are in the direction of better stability. We will revise this explanation in the camera-ready version.

---

> > ### Comment · Reviewer_yzUk · 2025-08-01
> >
> > Thank you for the reply. Looking forward to reading the incorporated explanations in the revised versions.

---

### Official Review · Reviewer_LzQr · 2025-07-14

**Clarity:** 4
**Significance:** 4
**Originality:** 3
**Rating:** 5
**Confidence:** 5

**Summary:**

This paper addresses the issue of temporal inconsistency that arises when single-frame models are applied to video data.
 The authors propose a simple yet effective stabilization module that can be inserted into pre-trained frame-based networks without modifying their parameters, enabling them to perform reliably on video tasks.

The proposed method introduces a lightweight stabilizer that operates on both intermediate features and final outputs, improving temporal stability and robustness to input corruptions (e.g., noise, compression, missing regions).
The method also leverages a unified accuracy–stability–robustness loss, and the authors provide theoretical analysis with the oracle bound and collapse bound to prevent over-smoothing and ensure well-behaved training.

Extensive experiments are conducted on a variety of tasks—image enhancement (HDRNet), video denoising (NAFNet), and depth estimation (Depth Anything v2)—demonstrating that the proposed stabilizers can significantly improve stability and robustness without sacrificing task accuracy, and in some cases even improving it.

Importantly, the stabilizers are causal and memory-efficient, making them well-suited for real-time and latency-sensitive applications.
 By avoiding the need to design task-specific video architectures, this work provides a practical and general-purpose framework for adapting existing frame-based models to video inference with minimal cost and effort.

**Questions:**

In this paper, the proposed method is evaluated by augmenting image-based baselines for various tasks.
However, for real-world video applications, it would be more appropriate to compare against video-specific baselines.
The paper would benefit from an analysis of how the proposed stabilizer performs when applied to existing video-based architectures.
For example, it would be interesting to see how much temporal consistency improves when applied to a video denoising model such as TurtleNet (NeurIPS 2024).
Including such comparisons would further strengthen the practical relevance of the work.

**Ethical Concerns:**

["NO or VERY MINOR ethics concerns only"]

**Final Justification:**

The proposed rebuttal addresses all of my concerns. Thank you to the authors for the response. I will keep my original rating for the final recommendation.

**Limitations:**

N/A - no known limitations

**Quality:**

3

**Strengths And Weaknesses:**

[Strengths]

The paper proposes a simple yet effective module for improving temporal consistency in video inference.
The approach is validated across multiple tasks, including image enhancement, denoising, and depth estimation, demonstrating consistent temporal stability improvements.
The qualitative video results provide strong evidence for the practical effectiveness of the proposed stabilizer in real-world scenarios.

[weaknesses]

The proposed stabilizer is essentially a combination of EMA and controller mechanisms, representing a well-engineered integration of existing ideas rather than a fundamentally novel concept.
As such, the contribution of the paper lies more in its practicality and generality, while its theoretical or architectural novelty is relatively limited.

Another weakness is that performance degrades under extreme noise conditions, especially with spatial fusion, which may limit robustness in harsh environments.

---

> ### Author Rebuttal · Authors · 2025-07-31
>
> We thank the reviewer for their feedback and respond to their comments below.
>
> **Performance degradation under extreme noise.** This is a great question, also raised by reviewer pzRZ. We duplicate our response below to make the rebuttal self-contained.
>
> We have examined these results more closely. When applying the spatial fusion method under extreme noise, we see some blurring/ghosting artifacts after enough time has passed. These artifacts can look like hard edges "bleeding out" into the surrounding regions. We believe this is related to the use of a spatial fusion stabilizer on the output layer. Without this stabilizer, the network output is biased toward the current input frame. However, adding a spatial fusion stabilizer to the output provides an independent mechanism for information to flow between pixels, thereby weakening this bias.
>
> Because we only see these artifacts after some time has passed, the problem may be resolved by training on longer sequences. We ran a quick experiment to test this idea. We trained the spatial fusion stabilizer under extreme noise on sequences of length 8 (the default in our past experiments) and 16. For length 16, we reduced the total number of training iterations by half, to compensate for the greater number of frames in each iteration. We evaluated these models on the full validation set (containing very long sequences). Training with length 8 gave validation PSNR 22.70 and instability 11.04, whereas training with length 16 gave PSNR 23.80 and instability 10.16. We expect this trend of improvement to hold as we further increase the training sequence length.
>
> We will incorporate this new experiment, sample results, and the above discussion in the supplement of our camera-ready version.
>
> **Comparison against video-specific baselines.** Thanks for the suggestion - we agree that the paper would be strengthened by comparisons against video-specific baselines, especially for tasks like denoising. Although we do not have time to add these comparisons now during the rebuttal, we intend to add at least one such baseline for the camera-ready version.
>
> We would also like to point out that our method is potentially complementary to many video-based models. Consider, for example, FastDVDNet, a video denoising model that maps five noisy input frames to one centrally-aligned denoised frame. To predict a denoised video, the model is applied in a sliding-temporal-window fashion to the input. Therefore, on each invocation of the model, the time offset of each output or activation value shifts forward by one frame. This mirrors the single-frame models we consider in the paper, where the activations also shift forward by one frame on each invocation. For other models which operate on distinct (non-overlapping) chunks of frames, our objective simply changes from "temporal consistency between frames" to "temporal consistency between chunks." The effectiveness of our method in this case would likely depend on the duration of the chunks - as chunks become longer the temporal correlation between them will tend to fall.

---

> > ### Comment · Reviewer_LzQr · 2025-08-03
> >
> > Thanks for your rebuttal. You have addressed all of my concerns. I look forward to seeing the final manuscript.

---

### Official Review · Reviewer_TYGq · 2025-07-19

**Clarity:** 3
**Significance:** 3
**Originality:** 3
**Rating:** 4
**Confidence:** 3

**Summary:**

The authors propose a novel method for stabilizing frame-based video inference, enhancing its robustness to temporal inconsistencies and corruptions. The approach is task-agnostic, model-agnostic, and lightweight—capable of being applied on top of pre-trained models without requiring retraining. It is also causal, meaning that predictions for later frames are informed by those of earlier frames.

**Questions:**

Why do the authors not compare against the consistent video enhancement that they cite in section 2? Namely, references [4,25,29,67,70].
How would this perform in transformer architectures? Is it limited only to convolutional architectures? If that was the case, it would be a significant limitation.

**Ethical Concerns:**

["NO or VERY MINOR ethics concerns only"]

**Final Justification:**

The authors have addressed my concerns.

**Limitations:**

Yes.

**Quality:**

2

**Strengths And Weaknesses:**

Strengths:
* The theoretical foundation is solid. The authors have developed their method, and the maths behind are explained soundly and clearly.
* The method is indeed lightweight, and the training procedure seems straightforward. In principle, it should be perfectly reproducible with the details provided in the manuscript.
* The method is adaptable for any architecture, although the authors only test it in convolutional layers.

Weaknesses:
* The experimental evaluation is very limited: the authors provide an ablation study without comparing against any existing SOTA solutions.
* The experiments are performed only in convolutional architectures. How would this perform in a transformer architecture?

---

> ### Author Rebuttal · Authors · 2025-07-31
>
> We thank the reviewer for their feedback and respond to their comments below.
>
> **Comparisons with consistent video enhancement.** Per the reviewer's suggestion, we have set up the author-published code for Bonneel et al. and have run their method on the NFS Laplacian task. We are currently seeing severe artifacts in the output of this method; we believe this is due to a misconfiguration and will continue to work on this throughout the discussion period. We will provide these results if we are able to obtain a valid comparison.
>
> There are several key conceptual distinctions between our method and these prior works. First, unlike consistent video enhancement methods, our approach does not require that the output is a natural video. This flexibility is evidenced by our results for other tasks, such as depth estimation. Since the submission we have also performed experiments for semantic segmentation, which we will include in the camera-ready version.
>
> Second, we consider not only stability, but also robustness, which allows our method to work even when the input video contains various corruptions (see Figure 6, Table 1, and Section 6.3 of our main paper). For example, the method of Bonneel et al. determines the amount of temporal smoothness to enforce by looking at low-level input cues, meaning it cannot smooth the output correctly in situations where the input video contains unstable corruptions.
>
> **Compatibility with transformer architectures.** Our method is indeed compatible with transformer architectures. When dealing with transformer layers, care must be taken to ensure that the controller output has the correct shape: a 1D list of tokens, rather than a 2D feature map. This can be achieved in several ways: by applying the backbone after an existing input tokenization, by adding a tokenization layer within the backbone, or by tokenizing the output of the controllers. In all of these cases, the controller layers after the tokenization should be switched from convolutions to MLP layers or self-attention blocks.

---

> > ### Comment · Reviewer_TYGq · 2025-08-08
> > **Comments addressed**
> >
> > The authors have addressed my comments, and, considering the points raised by my fellow reviewers and the responses of the authors, I have updated my rating.

---

### Official Review · Reviewer_pzRZ · 2025-07-22

**Clarity:** 4
**Significance:** 4
**Originality:** 3
**Rating:** 5
**Confidence:** 3

**Summary:**

This paper talks about how to fix the flickering and shakiness we see when using normal image neural networks on videos. These models process each frame by itself, so the output is not smooth. The authors made a general method to fix this. They created small "stabilization adapters" that you can plug into any pre-trained model without needing to retrain the whole thing. These adapters are controlled by a small neural network that looks at recent frames to decide how much to smooth the image features. They also introduce a new loss function to balance accuracy and stability, and they did some math to prove when it will work properly. They show their method works well for image enhancement, denoising, and depth estimation, making the videos smoother and more robust to noise and other problems

**Questions:**

1. Your spatial fusion method performed very poorly on the NFS extreme noise setting. You suggest it's an "unexpected attractor state" and maybe increasing τ in training could fix it. Have you tried this? Can you give more idea on why this happens? Why only for spatial fusion and not the normal controlled stabilizer?
2. For the video denoising part, the "Internal EMA" baseline was very bad because it tried to smooth the noise residual. This is a very interesting finding. Does it mean your learned controllers are smart enough to learn not to smooth these layers? It would be great if you could show this, for example by plotting the β values that the controller predicts for these layers. I would expect them to be close to 1.
3. You tested against adversarial attacks using I-FGSM. This is good. Did you consider testing against other types of attacks? For example, attacks that are not based on gradient, to see if the stabilizer is still robust

**Ethical Concerns:**

["NO or VERY MINOR ethics concerns only"]

**Final Justification:**

After author feedback, i would keep my rating of accept.

**Limitations:**

Yes. The authors included a limitations paragraph in the discussion

**Paper Formatting Concerns:**

I did not find any major formatting issues. The paper looks good.

**Quality:**

4

**Strengths And Weaknesses:**

Strengths:

1. The idea is very practical and useful. You don't need to build and train huge video models from scratch. You can just take an existing, powerful image model and add these lightweight adapters to make it work well on video. This will save lot of time and money for many people.
2. The paper has good theory. The unified loss and the "oracle bound" and "collapse bound" are very clever. It gives a clear reason for why we should pick a certain value for the hyperparameter. This is much better than just guessing.
3. The experiments are very thorough. They test the idea on three different tasks (HDRNet, NAFNet, Depth Anything) which shows that the method is general and not just for one problem. They also test against many types of image corruption, which is great.
4. The results are impressive. The method makes the video output much more stable, and in many cases, it also improves the PSNR or other metrics, especially when the input is corrupted. The video demos also show the improvement very clearly.

Weaknesses:
1. he theory has some limitations. The bounds they proved only work for simple loss functions that are a norm. For more complex loss functions used today, the theory might not hold. The authors do mention this, which is good.
2. The spatial fusion method, which is their best one, failed on the extreme noise experiment. They said it might be some "attractor state" but did not fully explain it. This is a point of failure for the method.
3. They don't report error bars or run experiments multiple times. They say it's because of compute cost, which is understandable, but it makes the results less statistically strong.

---

> ### Author Rebuttal · Authors · 2025-07-31
>
> We thank the reviewer for their feedback and respond to their comments below.
>
> **Lack of error bars.** We agree that adding error bars would strengthen the experimental results. Before camera-ready, we will run one set of experiments (a particular model and task configuration) multiple times to generate confidence bounds.
>
> **Analysis of failure cases with spatial fusion.** This is a great question, also raised by reviewer LzQr. We duplicate our response below to make the rebuttal self-contained.
>
> We have examined these results more closely. When applying the spatial fusion method under extreme noise, we see some blurring/ghosting artifacts after enough time has passed. These artifacts can look like hard edges "bleeding out" into the surrounding regions. We believe this is related to the use of a spatial fusion stabilizer on the output layer. Without this stabilizer, the network output is biased toward the current input frame. However, adding a spatial fusion stabilizer to the output provides an independent mechanism for information to flow between pixels, thereby weakening this bias.
>
> Because we only see these artifacts after some time has passed, the problem may be resolved by training on longer sequences. We ran a quick experiment to test this idea. We trained the spatial fusion stabilizer under extreme noise on sequences of length 8 (the default in our past experiments) and 16. For length 16, we reduced the total number of training iterations by half, to compensate for the greater number of frames in each iteration. We evaluated these models on the full validation set (containing very long sequences). Training with length 8 gave validation PSNR 22.70 and instability 11.04, whereas training with length 16 gave PSNR 23.80 and instability 10.16. We expect this trend of improvement to hold as we further increase the training sequence length.
>
> We will incorporate this new experiment, sample results, and the above discussion in the supplement of our camera-ready version.
>
> **Smoothing of the noise residual.** To answer this question, we look at the "learned EMA" configuration, which is the same as the "internal EMA" variant except that we train the decay parameters (with one parameter per feature channel). We consider the NAFNet NFS model trained under moderate noise with $\lambda = 0.2$. The table below shows the mean of the decay logits for each layer, along with the feature dimension to give a sense for the level of visual abstraction. Larger logit values correspond to beta near one (minimal smoothing). The last row is for the stabilizer on the model output.
>
> By far, the most aggressive stabilization is applied to the model output. Within the residual-prediction backbone, we see the most stabilization in the intermediate layers. These intermediate layers correspond to higher-level features that should be more stable than either the noisy input image or the residual prediction. Note that although the mean logit values are quite high, there is significant variation across channels, with many channels having values in the 0-1 range. This suggests that some channels are much more amenable to stabilization, and supports the use of training over hand-tuning the stabilizer decay parameters.
>
> We will include this new experiment and our discussion in the supplement.
>
> | Module name | Feature dimension | Logit mean |
> | --- | --- | --- |
> | module.encoders.0.0 | 32 | 12.51 |
> | module.encoders.0.1 | 32 | 12.03 |
> | module.encoders.1.0 | 64 | 10.37 |
> | module.encoders.1.1 | 64 | 9.91 |
> | module.encoders.2.0 | 128 | 9.41 |
> | module.encoders.2.1 | 128 | 9.50 |
> | module.encoders.2.2 | 128 | 9.75 |
> | module.encoders.2.3 | 128 | 9.33 |
> | module.encoders.3.0 | 256 | 8.14 |
> | module.encoders.3.1 | 256 | 8.56 |
> | module.encoders.3.2 | 256 | 8.62 |
> | module.encoders.3.3 | 256 | 8.61 |
> | module.encoders.3.4 | 256 | 8.61 |
> | module.encoders.3.5 | 256 | 8.63 |
> | module.encoders.3.6 | 256 | 8.64 |
> | module.encoders.3.7 | 256 | 8.24 |
> | module.middle_blks.0 | 512 | 5.55 |
> | module.middle_blks.1 | 512 | 5.64 |
> | module.middle_blks.2 | 512 | 5.29 |
> | module.middle_blks.3 | 512 | 5.38 |
> | module.middle_blks.4 | 512 | 5.33 |
> | module.middle_blks.5 | 512 | 5.26 |
> | module.middle_blks.6 | 512 | 5.32 |
> | module.middle_blks.7 | 512 | 5.40 |
> | module.middle_blks.8 | 512 | 5.32 |
> | module.middle_blks.9 | 512 | 5.31 |
> | module.middle_blks.10 | 512 | 5.16 |
> | module.middle_blks.11 | 512 | 4.52 |
> | module.decoders.0.0 | 256 | 7.63 |
> | module.decoders.0.1 | 256 | 6.44 |
> | module.decoders.1.0 | 128 | 7.85 |
> | module.decoders.1.1 | 128 | 7.40 |
> | module.decoders.2.0 | 64 | 8.71 |
> | module.decoders.2.1 | 64 | 7.45 |
> | module.decoders.3.0 | 32 | 10.33 |
> | module.decoders.3.1 | 32 | 10.19 |
> | module | 3 | 1.88 |
>
> **Testing against other adversarial attacks.** This is a good suggestion, although so far we have not tried this. In general, we would expect non-gradient-based attacks to be weaker than gradient-based attacks, and therefore easier to defend.

---

> > ### Comment · Reviewer_pzRZ · 2025-08-05
> > **Satisfied with the rebuttal**
> >
> > Thanks for the rebuttal. Most of my concerns have been addressed.

---

### Author Response · Authors · 2025-08-06
**Author Follow-Up**

Thanks to the reviewers for the encouraging and constructive comments so far. Please let us know if there are any further questions we can address during the discussion period.

---

### Decision · Program_Chairs · 2025-09-17

**Decision:**

Accept (poster)

**Comment:**

The paper proposes a method for stabilization of frame-based video inference. The contributions of this paper have a practical character and likely to make an impact on related approaches, given the modularity and transferability of the trained adapters to different architectures. The experimental analysis is solid, being it performed on three different tasks and showing good results compared to other methods as well as other baselines. The authors have also discussed limitations (e.g. the results in highly-noisy cases) and provided clarifying arguments for the related doubts. All-in-all the paper provides strong contributions, with thorough experiments and is supported by a convincing rebuttal.